# Microsecond sub-domain motions and the folding and misfolding of the mouse prion protein

**Rama Reddy Goluguri[1], Sreemantee Sen[1], Jayant Udgaonkar[1,2]***

[1]National Centre for Biological Sciences, Tata Institute of Fundamental Research, Bengaluru, India; [2]Indian Institute of Science Education and Research, Pune, India

**Abstract** Protein aggregation appears to originate from partially unfolded conformations that are sampled through stochastic fluctuations of the native protein. It has been a challenge to characterize these fluctuations, under native like conditions. Here, the conformational dynamics of the full-length (23-231) mouse prion protein were studied under native conditions, using photoinduced electron transfer coupled to fluorescence correlation spectroscopy (PET-FCS). The slowest fluctuations could be associated with the folding of the unfolded state to an intermediate state, by the use of microsecond mixing experiments. The two faster fluctuations observed by PET-FCS, could be attributed to fluctuations within the native state ensemble. The addition of salt, which is known to initiate the aggregation of the protein, resulted in an enhancement in the time scale of fluctuations in the core of the protein. The results indicate the importance of native state dynamics in initiating the aggregation of proteins.

DOI: https://doi.org/10.7554/eLife.44766.001

## Introduction

Structural fluctuations in proteins are not only critical for function (*Henzler-Wildman and Kern, 2007*; *Yang et al., 2014*) and folding (*Wani and Udgaonkar, 2009*; *Malhotra and Udgaonkar, 2016*; *Frauenfelder et al., 1991*; *Bai et al., 1995*), but also have a role to play in protein misfolding and aggregation (*Scheibel, 2001*; *Elam et al., 2003*; *Chiti and Dobson, 2009*; *Singh and Udgaonkar, 2015a*). Thermal fluctuations can lead to the sampling of partially unfolded conformations, which may or may not be populated transiently on protein folding and unfolding pathways (*Matouschek and Fersht, 1993*; *Sridevi and Udgaonkar, 2002*), and which may be aggregation-prone (*Hamid Wani and Udgaonkar, 2006*; *Moulick and Udgaonkar, 2017*; *Booth et al., 1997*). It is important to determine whether aggregation-competent conformations are indeed formed on the protein (un)folding pathway, to determine their structures, and to define the timescales of the structural fluctuations that lead to their transient accumulation. This is particularly true in the case of the prion protein, because 85% of prion diseases occur in a sporadic manner, presumably through stochastic fluctuations in the native state of the protein. In this context, it is important to note that the prion protein has a structure that is highly dynamic, as reflected in its unusually high native state heat capacity (*Moulick and Udgaonkar, 2014*).

In the case of the prion protein, misfolding and aggregation are linked to several fatal neurodegenerative diseases (*Collinge, 2001*). The cellular form of the prion protein, PrP$^C$, is composed of an unstructured N-terminal region, and a structured C-terminal domain that is comprised of three α-helices (α1, α2, α3) and two short β-strands (β1, β2) (*Riek et al., 1996*). Prion disease is caused by the infectious scrapie form, PrP$^{Sc}$, which is multimeric, β-sheet rich, partially resistant to proteinase K digestion (*Caughey et al., 1991*), and is capable of forming amyloid fibrils (*Prusiner, 1998*). Spontaneous conversion of PrP$^C$ to PrP$^{Sc}$, appears to occur in 85% of prion diseases, but the nature and

***For correspondence:**
jayant@iiserpune.ac.in

**Competing interests:** The authors declare that no competing interests exist.

time scales of prion protein dynamics that dictate this conformational conversion are not well understood.

It has been suggested that sub-domain separation of β-strands, β1 and β2, along with helix1 (α1) from the core helices, α2 and α3, initiates the aggregation of the prion protein (*Eghiaian et al., 2007*) (*Figure 1*). The aggregation of pathogenic (*Singh and Udgaonkar, 2015a*) and stabilized (*Singh et al., 2014*) variants of mouse prion protein (moPrP) has also been shown to proceed through a similar sub-domain separation mechanism (*Singh and Udgaonkar, 2015b*). It is not known to what extent such sub-domain separation happens during the stochastic excursions within the N state ensemble to misfolding-prone conformational states, under equilibrium native-like conditions.

Prion protein dynamics has been probed extensively by hydrogen-deuterium exchange (HDX). Native state HDX studies carried out under conditions that facilitate prion misfolding and aggregation, identified partially unfolded forms (PUFs) in equilibrium with the N state, which have been implicated in the aggregation of the protein (*Moulick et al., 2015*; *Moulick and Udgaonkar, 2017*). The HDX studies did not, however, allow the temporal sequence of the structural transitions to be determined. Moreover, because the prion protein folds and unfolds within a millisecond (*Honda et al., 2015*), it was not possible to ascertain whether the PUFs observed in the native-state HDX studies are also populated on the folding/unfolding pathway of the protein. To fully understand the nature of native state fluctuations at equilibrium, it becomes necessary to also carry out kinetic studies of folding and unfolding in the microsecond time domain.

Studies of the folding and unfolding of several prion proteins have identified states intermediate in their structures to the structures of the native (N) and unfolded (U) states, whose accumulation has been linked to prion protein aggregation (*Apetri and Surewicz, 2002*; *Apetri et al., 2006*; *Chen et al., 2011*; *Honda et al., 2015*). Little is, however, known about the structures of these (un) folding intermediates, and it is not known whether misfolding originates directly from the identified intermediates. This is important to demonstrate because under native-like conditions, proteins may

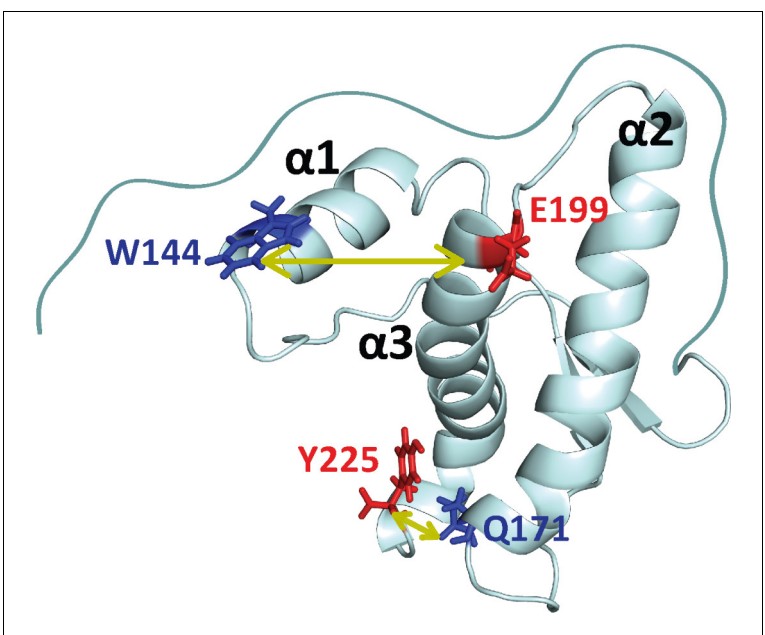

**Figure 1.** Structure of moPrP. Full length moPrP (23-231) is shown (PDB ID: 1AG2). The two distances, W144/C199 and W171/C225, across which the structural dynamics were monitored by PET-FCS are indicated by yellow arrows. For each distance, the position of the quencher (Trp residue) is indicated in blue, and the position where a Cys residue was introduced for Atto655 labeling, is colored red. The three α-helices (α1, α2, α3) are indicated.
DOI: https://doi.org/10.7554/eLife.44766.002
The following figure supplement is available for figure 1:

**Figure supplement 1.** Far-UV CD spectra of different labeled and unlabeled mutant variants of moPrP.
DOI: https://doi.org/10.7554/eLife.44766.003

sample not only bonafide (un)folding intermediates, but also partially unfolded forms that may not be *en route* to the globally unfolded state (*Sridevi and Udgaonkar, 2002*; *Matouschek and Fersht, 1993*). Ascertaining whether the aggregation prone intermediate forms on or off the folding pathways has a direct bearing on the development of strategies to combat aggregation because on-pathway intermediate forms could be expected to be longer-lived than transiently sampled native-like states.

The FCS methodology has been used extensively to study the nature of the fluctuations of the native state, and of the conformations transiently sampled under native equilibrium conditions (*Elson and Magde, 1974*; *Maiti et al., 1997*; *Haupts et al., 1998*; *Chattopadhyay et al., 2005*). Fluorescence quenching of oxazine fluorophores by photoinduced electron transfer (PET) when combined with FCS, results in the dynamics of the protein appearing as fluorescence fluctuations. PET-FCS has been used extensively to measure the dynamics of proteins in the native (*Doose et al., 2009*; *Ries et al., 2014*; *Neuweiler et al., 2009*), unfolded (*Sherman and Haran, 2011*; *Daidone et al., 2010*) and intermediate states (*Neuweiler et al., 2010*).

In the current study, PET-FCS measurements have been used together with kinetic measurements of the folding and unfolding of moPrP, to understand the dynamics of the mouse prion protein. PET probes were placed at two different locations to probe the dynamics between helices α1-α3 and α2-α3, in order to understand the role of sub-domain separation in initiating the aggregation of the prion protein.

Conformational dynamics across helices α1 and α3, and across helices α2 and α3, were monitored separately in two mutant variants of the protein designed for PET-FCS measurements (*Figure 1*). In each mutant variant, the dynamics leading to the quenching of an Atto 655 moiety introduced into the protein, by a suitably placed Trp residue, when the two come within contact quenching distance (<1 nm) (*Vaiana et al., 2003*), was measured as fluctuations in the Atto 655 fluorescence. Autocorrelation functions (ACFs) for both the distances show three exponential processes, arising from sub-millisecond dynamics in the protein. The slow exponential process was found to represent the dynamics of folding from the U state to an intermediate state (I), by comparing the kinetics obtained from PET-FCS to the kinetics of folding of the Atto 655-labeled protein. The two faster exponential processes could be attributed to fluctuations in the native state. Overall, it is shown that the native state dynamics of moPrP result in global unfolding *via* an intermediate. The addition of salt which is known to initiate aggregation, resulted in an enhancement in the time scale of fluctuations in the core of the protein. The current study detects fluctuations in the native state that lead to the sampling of aggregation-competent conformations of the protein.

## Results

### moPrP constructs for PET-FCS experiments

In the current study, Trp and Cys residues were engineered at different positions in moPrP in such a way that they would come in contact due to relative motion of helices in the native state of the protein. In the case of W144/C199-Atto moPrP, W144 was used as the PET quencher of the Atto 655 moiety attached to the Cys residue at position 199, to report on the dynamics across helices α1 and α3 (*Figure 1*). In the case of W171/C225-Atto moPrP, W144 was mutated to a Phe residue, and a Trp residue was introduced at position 171. A Cys residue was incorporated at position 225, and this construct (W171/C225) is expected to report on the dynamics between helices α2 and α3 (*Figure 1*). Labeling with Atto 655 did not result in any alteration in the secondary structure of the protein, as indicated by the far-UV circular dichroism measurements (*Figure 1—figure supplement 1*).

### PET-FCS can be used to measure the rate constants of conformational fluctuations

In the current study, PET-FCS was used to determine the kinetics of protein fluctuations that enable the formation of quenching-competent dye-Trp complexes in which quenching of the fluorescence of the oxazine dye by PET from the Trp side-chain, can occur. It is known that such quenching occurs primarily by a static quenching mechanism (*Sauer and Neuweiler, 2014*). After the formation of productive dye-Trp complexes, in which the dye and quencher are in van der Waals contact (*Vaiana et al., 2003*), PET from the Trp side-chain leads to the quenching of the dye fluorescence,

resulting in essentially a non-fluorescent ground state complex. Since PET occurs within a few pico-seconds (*Lakowicz, 1999*; *Zhong and Zewail, 2001*; *Doose et al., 2005*), and because dissociation of the dye-Trp complex is much slower (*Sherman and Haran, 2011*; *Soranno et al., 2017*), the observed rate constant of the overall quenching reaction reflects the rate constant for the formation of the dye-Trp complex.

It is known that PET occurs by a distance-dependent quenching mechanism, in which the electron transfer rate constant decreases exponentially with an increase in the distance separating the fluorophore from the Trp side-chain (*Lakowicz, 1999*). This would mean that if there were a distribution of fluorophore-Trp distances in the protein molecules comprising the native state ensemble, the observed transfer rate constant in PET would represent a rate constant averaged across all distances, as is the case for the rate constant of energy transfer observed in FRET measurements (*Schuler and Hofmann, 2013*). However, in PET, the probability of electron transfer decreases asymptotically to near zero within less than a nanometer of separation; and the quenching distance, above which electron transfer cannot occur, is about 0.5 nm (*Lewis et al., 1997*; *Vaiana et al., 2003*), which corresponds to van der Waals contact between the dye and Trp side-chain. Since diffusive fluctuations over such small distances will occur on the nanosecond time scale, it is very unlikely that the ~1 μs and slower dynamics observed in the current study are the result of distance-dependent ensemble-averaging of the quenching process. It is because quenching occurs only when the dye and Trp side chain are in van der Waals contact in dye-Trp complexes, that it has been referred to as an on-off process (*Schuler and Hofmann, 2013*; *Sauer and Neuweiler, 2014*).

In studies of the PET quenching of free dye by free tryptophan, it was observed that the observed quenching rate constant was 2 to 3 times slower than the bimolecular collision rate constant for a diffusion-controlled reaction (*Soranno et al., 2017*). It appeared that only one in two or three collisions led to the formation of a productive complex in which the dye and Trp possessed the appropriate geometries of stacking of the indole ring of the Trp with the aromatic groups of the dye, which enabled PET to occur (*Haenni et al., 2013*; *Doose et al., 2005*). When the dye and Trp are attached to a protein chain, as in the current study, then the rate constant at which quenching will be observed will depend on the rate constant of the protein fluctuation/conformational change that allows the dye and Trp to associate to form a quenching-competent complex. In the current study, the observed kinetics of quenching, which reflects the kinetics of formation of a quenching-competent complex, have been used as a measure of the kinetics of conformational dynamics of the protein.

## moPrP shows three reaction kinetic phases in the sub-millisecond time scale

The ACFs of both W144/C199-Atto moPrP and W171/C225-Atto moPrP, showed three exponential components, in addition to the diffusion component (*Figure 2a,b*). The ACFs of both the PET constructs and Trp-less control proteins fit well to a model which had three exponential components, as seen by the residuals to the fits (*Figure 2—figure supplement 1*). The values of the parameters obtained from the fits to the Trp containing proteins are shown in *Table 1*. The total amplitude of the exponential components for W144-C199-Atto moPrP was higher than that observed for W171/C225-Atto moPrP. The time constants and amplitudes of the exponential processes were found to be independent of the excitation power used in the FCS experiments (*Supplementary file 1*). Two control Trp-less proteins in which all the Trp residues were mutated to Phe, and a Cys residue was placed at either position 199 (control for W144/C199-Atto moPrP) or at position 225 (control for W171/C225-Atto moPrP), were used to confirm that the dynamics observed in the PET-FCS experiments were indeed due to PET quenching of Atto 655 fluorescence by the Trp residue. The Trp-less control proteins showed negligible exponential components, confirming that these components originate from the quenching of the fluorescence of the Atto moiety attached to the protein by the Trp residue in the Trp-containing proteins (*Figure 2a,b*). The very minor exponential components seen for the control proteins can be attributed to the environment sensitivity of the Atto dye attached to the protein (*Ries et al., 2014*). Purified samples that had been buffer-exchanged further to remove any free dye present by an additional 1000-fold, did not show any change in the exponential components, ruling out the possibility of the presence of free dye in the samples, that could have resulted in the appearance of faster components in the ACFs.

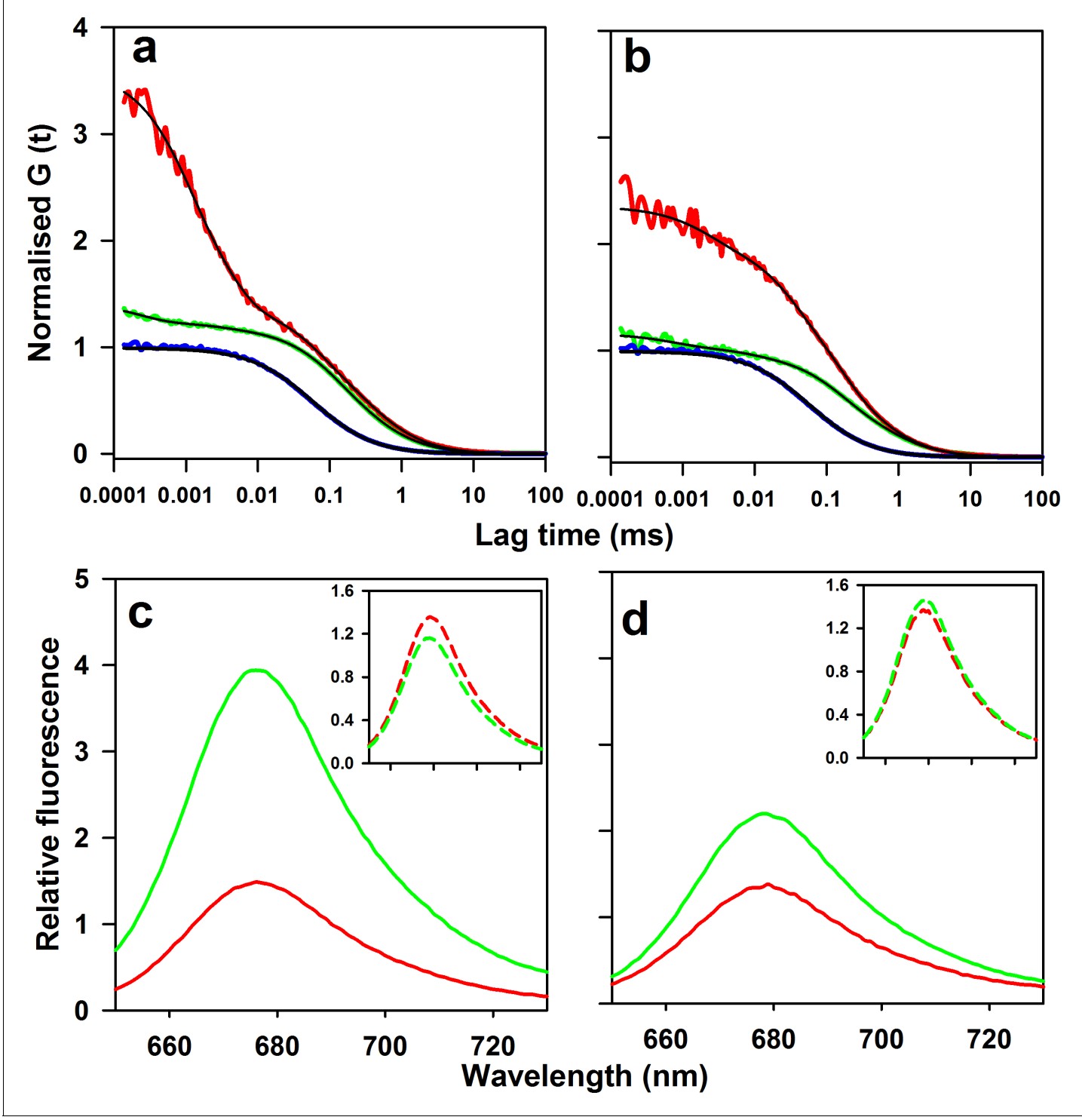

**Figure 2.** Microsecond dynamics of moPrP at pH 4 monitored by PET-FCS. (**a**) The autocorrelation function (ACF) of W144/C199-Atto moPrP is shown in red. The ACF of free Atto dye, acquired under identical conditions, is shown in blue. The ACF of the control Trp-less variant, C199-Atto moPrP, is shown in green. The fits to appropriate models are shown in black for each ACF. (**b**) The ACF of W171/C225-Atto moPrP, free Atto dye and the Trp-less control protein C225-Atto moPrP are shown in red, blue and green, respectively. The fits to appropriate models are shown in black for each ACF. The values of $\tau_d$ for free Atto dye, W144/C199-Atto moPrP and W171/C225-Atto moPrP are 66 ± 5, 304 ± 22 and 316 ± 30 µs. (**c**) Atto fluorescence emission spectra of W144/C199-Atto moPrP (red) and the corresponding Trp-less control protein C199-Atto moPrP (green) at pH 4, are shown. (**d**) Atto 655 fluorescence emission spectra of W171/C225-Atto moPrP (red) and corresponding Trp-less control protein C225-Atto moPrP (green) at pH 4, are shown.

*Figure 2 continued on next page*

*Figure 2 continued*

In panels c and d, the spectra of the native proteins are shown as continuous lines; the insets show the spectra of the unfolded proteins (in 6 M Urea) as dashed lines (inset). The spectra were normalized to the fluorescence of the Trp-containing protein at 690 nm.

DOI: https://doi.org/10.7554/eLife.44766.004

The following source data and figure supplements are available for figure 2:

**Source data 1.** ACFs and fluorescence spectra of moPrP mutant variants, control Trp-less proteins and dye.

DOI: https://doi.org/10.7554/eLife.44766.011

**Figure supplement 1.** Residuals of fits to the ACF to *Equation 1* in main text.

DOI: https://doi.org/10.7554/eLife.44766.005

**Figure supplement 1—source data 1.** Residuals of fits to the ACF.

DOI: https://doi.org/10.7554/eLife.44766.006

**Figure supplement 2.** ACF of W144/C199-Atto moPrP in 6 M urea at pH 4.

DOI: https://doi.org/10.7554/eLife.44766.007

**Figure supplement 2—source data 1.** ACF of W144/C199-Atto moPrP in6Murea at pH 4.

DOI: https://doi.org/10.7554/eLife.44766.008

**Figure supplement 3.** Equilibrium unfolding transition of Atto655-labeled mutant variants of moPrP.

DOI: https://doi.org/10.7554/eLife.44766.009

**Figure supplement 3—source data 1.** Equilibrium unfolding transition of Atto655-labeled mutant variants of moPrP.

DOI: https://doi.org/10.7554/eLife.44766.010

The advantage of using the PET-FCS methodology to monitor the native state dynamics of the prion protein is that the results are unlikely to be complicated by potential aggregation. These experiments were carried out at a final protein concentration of 2–20 nM; at these concentrations the protein is not expected to aggregate within the time scale of measurement. This was directly confirmed by experimentally determining the value of the diffusion coefficient, which was found to be $90 \pm 10$ $\mu m^2/s$ for the different variants. The hydrodynamic radius calculated from the diffusion coefficient obtained from the PET-FCS measurements was 2.8 nm, which matches well with the $R_H$ measured by dynamic light scattering experiments in earlier studies (*Jain and Udgaonkar, 2008*). Another advantage of using PET FCS for measuring the conformational dynamics of proteins, is that the dynamics can be directly monitored, as the actual PET process is very fast and happens on the picoseconds time scale.

**Table 1.** Parameters obtained from the ACFs of W144/C199-Atto moPrP and W171/C225-Atto moPrP.

The parameters defining the dynamics in the absence of added salt were obtained from the fits to the data shown in *Figure 2a and b*. The errors shown are the standard deviations determined from measurements made in at least two independent experiments.

|  | W144/C199-Atto moPrP | | W171/C225-Atto moPrP | |
|---|---|---|---|---|
|  | pH 4 | pH 4, 150 mM NaCl | pH 4 | pH 4, 150 mM NaCl |
| $K_1$ | $1.2 \pm 0.1$ | $0.9 \pm 0.2$ | $0.3 \pm 0.07$ | $0.65 \pm 0.08$ |
| $K_2$ | $0.8 \pm 0.4$ | $1.3 \pm 0.2$ | $0.4 \pm 0.1$ | $0.3 \pm 0.1$ |
| $K_3$ | $0.4 \pm 0.02$ | $0.2 \pm 0.02$ | $0.65 \pm 0.3$ | $0.25 \pm 0.1$ |
| $\tau_1$ (µs) | $1.1 \pm 0.3$ | $0.8 \pm 0.1$ | $1 \pm 0.8$ | $0.5 \pm 0.06$ |
| $\tau_2$ (µs) | $5 \pm 2$ | $3 \pm 0.2$ | $19 \pm 3$ | $4 \pm 1$ |
| $\tau_3$ (µs) | $80 \pm 11$ | $61 \pm 28$ | $122 \pm 11$ | $89 \pm 24$ |
| $\tau_D$ (µs) | $306 \pm 26$ | $321 \pm 20$ | $277 \pm 3$ | $323 \pm 20$ |

DOI: https://doi.org/10.7554/eLife.44766.012

## Structural fluctuations within the native state ensemble give rise to the faster exponential components in the PET-FCS experiments

The Cys and Trp residues in the two PET constructs were engineered at positions that are expected to come in contact with each other due to motions in the N state of the protein (*Figure 1*). Such a strategy has been used previously to study the native state dynamics of the ionotropic glutamate receptor (*Jensen et al., 2011*), spider silk protein (*Ries et al., 2014*) and Hsp 90 (*Schulze et al., 2016*). The quenching of the Atto 655 dye fluorescence by PET from the Trp residue in the two PET constructs will result in a fraction of molecules existing in non-fluorescent states. Consequently, the fluorescence emission intensity of the Trp-containing PET construct will be lower than that of the corresponding Trp-less PET construct (*Figure 2c,d*). With the assumption that the quenched states are completely non-fluorescent, the relative fluorescence intensity of the Trp-containing protein compared to Trp-less PET construct directly gives the fraction of molecules in the native state ensemble, in which the Atto 655 fluorescence is not quenched by PET from a Trp side-chain.

The observation of two conformational fluctuations in the 0.5–20 µs time domain in PET-FCS measurements of the native state suggested that conformational fluctuations of fluorescent N in the native state ensemble lead to the sampling of two non-fluorescent states, N* and N**(see below). Conformational fluctuations on this time scale have been observed in the native state ensembles of other proteins (*Jensen et al., 2011*; *Ries et al., 2014*; *Schulze et al., 2016*). The fluorescence of the Atto 655 dye moiety in N* and N** is quenched by PET occurring in the transient dye-Trp complexes formed within them, which probably differ in the orientation of the aromatic rings of the dye and Trp with respect to each other (*Vaiana et al., 2003*). The kinetics of formation of N* and N**, as well as their structures, would be dictated by the nature of the protein fluctuations that lead to their formation. The equilibrium constants $K_1$, $K_2$ between the two dark states N* and N**, and the fluorescent N conformation, could be calculated from the PET-FCS data, again with the assumption that all molecules were fluorescent in the Trp-less control protein, and that a fraction of the molecules in the Trp-containing protein existed in a completely dark state. For each of the PET constructs, the value of $K_1 + K_2$ was found to be similar (*Table 2*), whether calculated from the fluorescence emission spectra (*Figure 2c,d*), or from the PET-FCS data (*Figure 2a and b*). This similarity suggested that the two faster exponential components in the ACFs arise from structural fluctuations within the native state ensemble of moPrP.

It is also possible that N is non-fluorescent and that N* and N** are fluorescent. The current study cannot distinguish between the possibilities, and the model with one fluorescent N state and the two non-fluorescent N* and N** states, has been chosen only because it seems to be the simpler alternative. It should be noted that the alternative possibility does not affect any of the conclusions drawn in the current study. In both cases the fluctuations arise from N sampling N* and N** in the native state ensemble.

In the case of the unfolded state, the Atto 655 fluorescence spectrum of the Trp-containing protein overlapped with that of the corresponding Trp-less protein, for both the PET constructs (insets of *Figure 2c,d*), which could be due either to the absence of PET quenching in the unfolded state, which is very unlikely, or to a decrease in the efficiency of the PET process at high concentration of denaturant, which is known to happen (*Sherman and Haran, 2011*; *Soranno et al., 2017*). Indeed,

**Table 2.** Comparison of equilibrium constants between the dark (N*+N**) states and the fluorescent (N) state obtained from fluorescence spectra and PET-FCS.

The ratio of the fluorescence intensity of the Trp-containing PET construct to that of the corresponding Trp-less control protein (*Figure 2c and d*), which is equal to $1 + K_1+K_2$, was used to determine $K_1 +K_2$ ($(N*+N**)/N)$). $K_1 +K_2$ was also determined as the sum of the amplitudes of the two faster exponential components observed in the ACFs (*Figure 2a and b*). It was assumed that in both N* and N**, the fluorescence of the Atto 655 dye moiety is fully quenched by PET. The errors shown represent the standard deviations determined from at least two independent experiments.

| | ($K_1 + K_2$) calculated from fluorescence spectra | ($K_1 + K_2$) obtained from ACFs of PET-FCS |
|---|---|---|
| W144/C199-Atto moPrP | 1.8 ± 0.2 | 2 ± 0.5 |
| W171/C225-Atto moPrP | 0.5 ± 0.2 | 0.7 ± 0.2 |

DOI: https://doi.org/10.7554/eLife.44766.013

the total amplitude of the exponential components seen in PET-FCS experiments carried out with W144/C199-Atto moPrP in 6 M urea, was more than 10-fold lower than the amplitude observed in the absence of urea (*Figure 2—figure supplement 2*).

Furthermore, Atto 655 fluorescence-monitored equilibrium unfolding experiments showed a 4-fold and a 2-fold enhancement in fluorescence upon unfolding for W144/C199-Atto moPrP and W171/C225-Atto moPrP, respectively (*Figure 2—figure supplement 3*). The increase in fluorescence upon unfolding observed for both the PET constructs can be explained by the release of quenching of Atto 655 fluorescence, which occurs in the native state of the protein. The fluorescence of W171/C225-Atto moPrP in the unfolded baseline was found to be less than that of W144/C225-Atto moPrP, and also had a very strong dependence on denaturant concentration (*Figure 2—figure supplement 3*). This could be because of the presence of residual structure in the unfolded state of W171/C225-Atto moPrP, due to the presence of the disulphide bridge between α2 and α3.

## The U↔I transition gives rise to the slow exponential component in the PET-FCS experiments

There are three possible explanations for the slowest exponential component observed in the PET-FCS experiments. One explanation is that it represents slow dynamics in the unfolded state. While some studies have suggested that unfolded state dynamics can occur in the microsecond time domain (*Waldauer et al., 2010*), other studies have reported unfolded dynamics in the nanosecond time domain (*Möglich et al., 2006*; *Hagen et al., 1996*; *Lapidus et al., 2000*; *Soranno et al., 2017*). Another explanation is that it represents slow dynamics in the native ensemble, or perhaps even the sampling of a partially unfolded intermediate by N. This explanation cannot be fully ruled out. A third explanation is that the slowest component represents a fluctuation which drives the folding of the U state to an early folding intermediate.

To determine which of these alternative explanations is correct, the folding/unfolding dynamics of W144/C199-Atto moPrP were characterized using microsecond mixing experiments, which were carried out using a custom-built continuous-flow setup with a mixing dead time of 37 μs (*Goluguri and Udgaonkar, 2016*). The unfolding (*Figure 3a*) and refolding (*Figure 3b*) kinetics were measured by monitoring the fluorescence of the Atto dye at 680 nm. The unfolding of W144/C199-Atto moPrP occurred in a single kinetic phase (*Figure 3a*) that accounted for the entire equilibrium unfolding amplitude of unfolding (*Figure 3c*). Extrapolation of the denaturant dependence of the unfolding rate constant to zero urea concentration, predicted an apparent unfolding rate constant of about 400 s$^{-1}$ in native conditions.

The refolding of W144/C199-Atto moPrP occurred in two kinetic phases (*Figure 3b*), whose total amplitude accounted for the entire equilibrium amplitude of folding (*Figure 3c*). The data fit well to a two-exponential equation, as seen by the random distribution of residuals as well as visual inspection of the fit through the data (*Figure 3—figure supplement 1*). The simplest explanation for the observation of two kinetic phases is that folding occurs *via* an U↔ I ↔ N mechanism, in which I is a folding intermediate. The rate constant of the fast kinetic phase was found not to depend on the total flow rate during mixing and observation (*Figure 3d*). The slow kinetic phase had a rate constant of ~500 s$^{-1}$ (*Figure 3d*), and accounted for about 33% of the total amplitude of the decrease in fluorescence seen for folding from U to N. It is straightforward to assign the fast kinetic phase to the U ↔ I step, and the slow kinetic phase to the I ↔ N step of the U↔ I ↔ N mechanism. It should be noted that the possibility that I is, instead, an off-pathway intermediate, cannot be ruled out, even though several previous kinetic studies of the folding of the mouse prion protein (*Honda et al., 2015*; *Moulick et al., 2019*) and of other mammalian prion proteins (*Apetri et al., 2006*; *Chen et al., 2011*) have implicated an on-pathway intermediate being formed on a time scale similar to that observed in this study. It should be noted that the observation that I like N is destabilized upon the addition of salt, suggests that the salt-perturbed native interaction is also present in I, which is also consistent with I being populated on the pathway from U to N.

*Figure 3d* shows that the dependence on denaturant concentration of the rate constant of the fast phase observed in the microsecond mixing experiments, extrapolates into the dependence on denaturant concentration of the rate constant of the slower exponential process observed in the PET-FCS measurements. It should be noted that the PET quenching efficiency, which reflects the probability of forming the quenching-competent complex from the encounter complex, can be suppressed by denaturant (*Doose et al., 2005*), but not at the low concentrations (<0.75 M) used in the

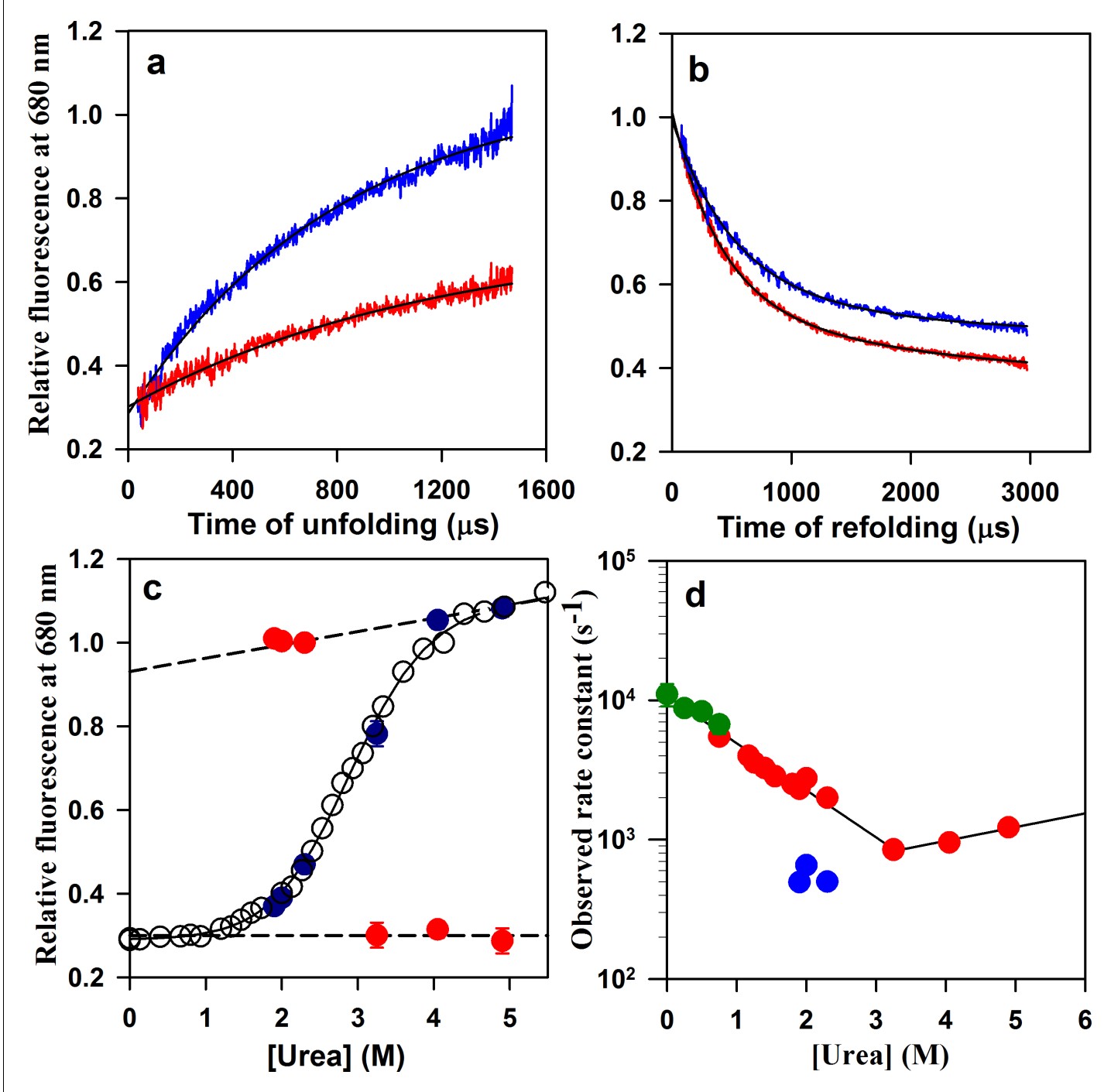

**Figure 3.** Folding/unfolding kinetics of W144/C199-Atto moPrP monitored by microsecond mixing experiments. (**a**) Unfolding kinetic traces at 3.25 M urea and 4.9 M urea are shown in red and blue, respectively. (**b**) Refolding kinetic traces at 1.9 M urea (red) and 2.3 M urea (blue) are shown. (**c**) A comparison of the equilibrium and kinetic fluorescence signal change upon folding/unfolding is shown. The equilibrium fluorescence transition is shown in black circles. The t = 0 values obtained from the kinetic traces are shown in red, and the t = ∞ points are shown in blue. The dashed lines are the linearly extrapolated native protein and unfolded protein baselines. (**d**) The apparent rate constants of the fast phase of folding and of unfolding (red symbols), obtained from the microsecond mixing experiments are plotted against urea concentration. Also plotted are the rate constants of the slow phase of folding (blue symbols). The microsecond mixing experiments for studying folding were carried out at two flow rates: 7 ml/min for Urea concentrations in the range of 1.2 to 1.9 M, and 3.5 ml/min for urea concentrations in the range of 1.9 to 2.3 M. The relaxation rate constants obtained from PET-FCS are shown in green. The black lines through the unfolding and refolding arms of the chevron are linear regression fits through the data points.

*Figure 3 continued on next page*

*Figure 3 continued*

DOI: https://doi.org/10.7554/eLife.44766.014

The following source data and figure supplements are available for figure 3:

**Source data 1.** Folding/unfolding kinetics of W144/C199-Atto moPrP monitored by microsecond mixing experiments.
DOI: https://doi.org/10.7554/eLife.44766.017
**Figure supplement 1.** Refolding kinetic trace of W144/C199-Atto moPrP at 1.9 M Urea concentration.
DOI: https://doi.org/10.7554/eLife.44766.015
**Figure supplement 1—source data 1.** Refolding kinetic trace of W144/C199-Atto moPrP at1.9MUrea concentration along with the residuals to fit to a single and double exponetial equation.
DOI: https://doi.org/10.7554/eLife.44766.016

current PET-FCS study (*Soranno et al., 2017*): the sum of the amplitudes of the exponential processes was found not to be affected by urea concentrations up to 0.75 M. It is also seen that the rate constant of the slow exponential process measured in 0.75 M urea by PET-FCS, matches the fast rate constant of folding in 0.75 M urea. It is important to note that the fast rate constants of folding in 1.25 to 2.5 M urea, where the quenching efficiency is suppressed, extrapolate to the rate constant of folding in 0.75 M urea (*Figure 3d*), where the quenching efficiency is unaffected by urea. This observation indicates that any suppression of the PET quenching efficiency that occurs at the high urea concentrations, does not significantly affect the time courses of folding at those urea concentrations.

The observation that the rate constant of the fast phase of folding observed in the microsecond mixing experiments was identical with the rate constant of the slow exponential process seen in the PET-FCS experiments, indicated that the same transition was being monitored in both measurements. This transition must be the U ↔ I transition, because the time constant (2 ms) of the I ↔ N transition is too slow to be measured by PET-FCS, given that $\tau_D$ has a measured value of ~300 μs. The structural segments spanning the two distances probed in the current study, showed similar timescales for the folding/unfolding dynamics when monitored by PET-FCS. It is important to note that the slowest exponential process seen in the PET-FCS experiments, represents a conformational fluctuation of the unfolded state at low concentrations of urea, which is not a random fluctuation, but which leads to the formation of a productive folding intermediate.

The amplitudes of the exponential quenching processes observed by PET-FCS correspond to the equilibrium constants between two conformations only when the fluorescence of one of the conformations is completely quenched (*Sauer and Neuweiler, 2014*; *Wu et al., 2016*). In the case of the two faster exponential processes corresponding to fluctuations between N and N*, and between N and N**, this criterion appears to be approximately met (see above). It is not possible, at the present time, to be certain that the fluorescence of the Atto moiety is completely quenched by the Trp residue, in I. There is, however, no data to support the dye-Trp complex being different in I than in the N ensemble. If complete quenching does not occur in the dye-Trp complex, then the amplitude of the slowest exponential process observed by PET-FCS would not correspond to the true equilibrium constant for the pre-equilibrium established between U and I during the fast kinetic phase of folding. In the absence of knowledge about the extent of fluorescence quenching in I, it is also difficult to determine the extent to which I is populated at the end of the fast phase.

## Folding scheme of moPrP

The results of the PET-FCS and microsecond mixing experiments can be explained by the reaction scheme shown in *Figure 4*. The faster dynamics are assumed to originate from fluctuations within the native state ensemble, leading to the sampling of non-fluorescent states N* and N**. The slowest dynamics arise from the transition between U and I.

## Salt affects both the fast and slow exponential processes observed by PET-FCS

Salt is known to trigger the aggregation of moPrP at pH 4 (*Jain and Udgaonkar, 2010*; *Singh and Udgaonkar, 2016*; *Sengupta and Udgaonkar, 2017*). In order to understand the effect of salt on the native state dynamics of moPrP, PET-FCS experiments were carried out in the presence of 150

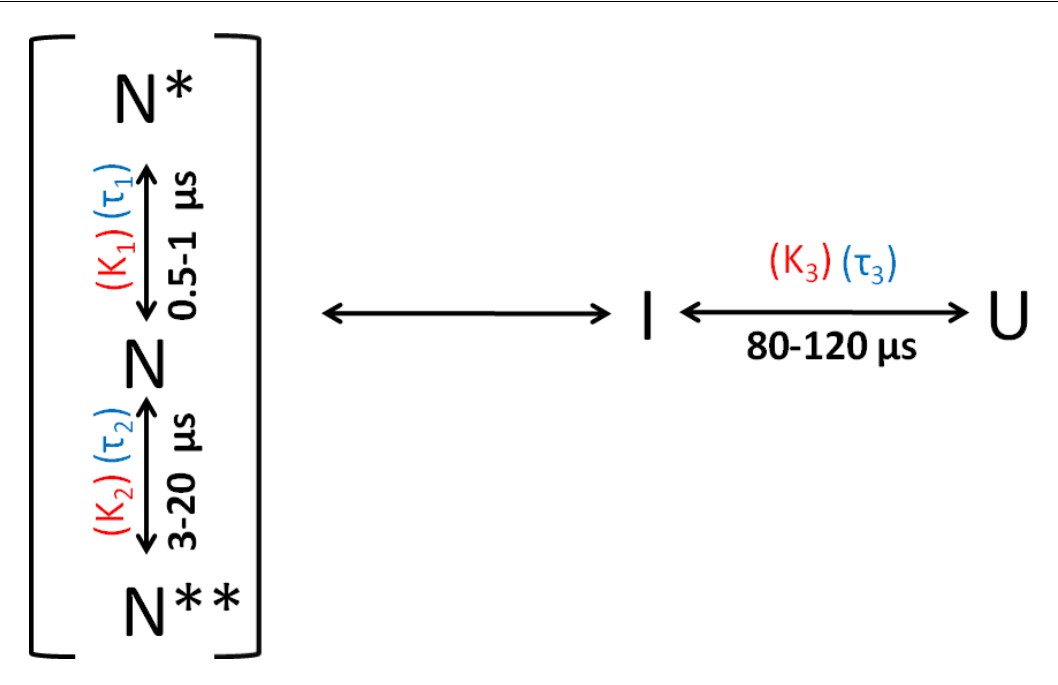

**Figure 4.** Folding scheme of moPrP at pH 4. The folding scheme is based on the data obtained from PET-FCS and microsecond mixing experiments. The faster exponential components in the ACFs of PET-FCS experiments have been attributed to native state dynamics. In non-fluorescent N* and N**, the Atto 655 fluorescence is quenched by the Trp-residue coming into contact to form dye-Trp complexes, during the N ↔ N* and N ↔ N** conformational transitions. The corresponding equilibrium constant ($K_{NN*}$) and time constant for the N to N* transition were obtained from the amplitude ($K_1$) and time constant ($\tau_1$) of the faster exponential component of the ACF, respectively. Similarly, the equilibrium constant ($K_{NN**}$) and the time constant for the N to N** transition were obtained from the $K_2$ and $\tau_2$ values of the ACFs. The slowest exponential component was found to correspond to the U ↔ I transition. The equilibrium constant ($K_{UI}$) and time constant for the U ↔ I transition were obtained from the amplitude ($K_3$) and time constant ($\tau_3$) of the slowest exponential component of ACF, respectively.
DOI: https://doi.org/10.7554/eLife.44766.018

mM NaCl. Thermodynamic parameters obtained from equilibrium unfolding experiments, and those calculated from the ACFs of PET-FCS experiments in the presence and absence of 150 mM NaCl, assuming the reaction scheme shown in *Figure 4*, are listed in *Table 3*. In the presence of salt, the combined amplitude (K1 +K2) of the two faster exponential processes was found to have slightly increased for W171/C225-Atto moPrP (*Figure 5a,b*). The time scale of the native state fluctuations was observed to become faster upon the addition of salt (*Table 1*). Equilibrium unfolding experiments carried out at pH four for W144/C199-Atto moPrP in the presence of salt indicated that the protein is destabilized by approximately 0.7 kcal/mol upon addition of salt (*Figure 5—figure supplement 1*). Both the amplitude and the time constant of the slowest exponential process was seen to decrease upon the addition of salt, at pH 4, for both the PET pairs (*Table 1*). The results of PET-FCS experiments at pH seven are shown in *Supplementary file 2*. The addition of salt was found to result in a similar change in the amplitudes and time constants of the exponential components at pH seven as well.

## Discussion

In the current study, the native state dynamics of moPrP were studied using the PET-FCS methodology. Under native conditions, moPrP displays conformational dynamics in three different time regimes. The two faster processes, in the 0.5–20 µs time regime, appear to correspond to structural fluctuations within the N state ensemble. The slowest dynamics appear to correspond to the transition of the protein between the unfolded state and an intermediate state.

**Table 3.** Effect of salt on thermodynamic parameters governing the dynamics of W144/C199-Atto moPrP.

The equilibrium constants were obtained from the amplitudes of ACFs obtained from PET-FCS experiments. Note that $K_{NN*}=N*/N$, $K_{NN**}=N**/N$ and $K_{UI} = I/U$. $\Delta G_{UI}$ is calculated from the value of $K_{UI}$. The total free energy change ($\Delta G_{(N+N*+N**)U}$) for unfolding was obtained from equilibrium unfolding experiments. The errors represent the standard deviations obtained from measurements made in at least two independent experiments.

| | W144/C199-Atto moPrP | |
| --- | --- | --- |
| | pH 4 | pH 4, 150 mM salt |
| $K_{NN*}$ ($K_1$) | 1.2 ± 0.1 | 0.9 ± 0.2 |
| $K_{NN**}$ ($K_2$) | 0.8 ± 0.4 | 1.3 ± 0.2 |
| $K_{UI}$ ($K_3$) | 0.4 ± 0.02 | 0.2 ± 0.02 |
| $\Delta G_{UI}$ (kcal/mol) | 0.5 ± 0.1 | 0.9 ± 0.1 |
| $\Delta G_{(N+N*+N**)I}$ (kcal/mol) | 4.4 ± 0.2 | 4.1 ± 0.2 |
| $\Delta G_{(N+N*+N**)U}$ (kcal/mol) | 3.9 ± 0.3 | 3.2 ± 0.3 |

DOI: https://doi.org/10.7554/eLife.44766.023

## moPrP folds in a similar manner at low and high denaturant concentrations

Conventionally, the folding/unfolding dynamics of proteins are studied by denaturing the protein using denaturants and then initiating the refolding reaction by rapid dilution of the denaturant. However, the mechanism of folding for a few proteins has been shown to be different, under native conditions, from what is expected from such refolding studies carried out in the presence of denaturant (*Sridevi and Udgaonkar, 2002*; *Matouschek and Fersht, 1993*). In some cases, denaturants have been shown to affect the cooperativity of the folding reaction and the ruggedness of the energy landscape (*Malhotra and Udgaonkar, 2015*; *Malhotra et al., 2017*; *Jethva and Udgaonkar, 2017*; *Moulick et al., 2019*). Hence, in light of the fact that denaturants can potentially modulate the folding landscape of a protein, it was important to study the folding pathway of the prion protein under native conditions.

The PET-FCS methodology is an ideal probe for monitoring the folding/unfolding dynamics of proteins that fold in the microsecond time regime, under native conditions. It has been employed successfully to study the folding reactions of a few fast folders, including the Trp-cage (*Neuweiler et al., 2005*), BBL (*Neuweiler et al., 2009*), Engrailed Homeodomain (*Neuweiler et al., 2010*) and spider silk protein (*Ries et al., 2014*). In these earlier studies, fluorescent and non-fluorescent (quenched) states could be identified as distinct states present on the folding pathway. In the current study, the slower exponential process observed by PET-FCS was identified by microsecond mixing folding experiments to arise from the U ↔ I folding transition, with U being the fluorescent state, and I being the non-fluorescent state. The observation that exponential extrapolation of the rate constants obtained from kinetic experiments of refolding at higher denaturant concentration, predicts the rate constant at zero denaturant (determined by PET-FCS) indicates that moPrP folds through same pathway in both the presence and absence of urea, although the possibility of the operation of multiple folding routes cannot be negated.

## The two faster PET-FCS fluctuations likely represent native state fluctuations

While the slow exponential process observed by PET-FCS could attributed to the U ↔ I step in the U↔ I ↔ N mechanism describing folding, the two faster exponential processes could arise from fluctuations in any of the three states, that is U, I or N. The U state of any protein is very dynamic. In the case of moPrP, the native state has been shown to be unusually dynamic (*Moulick and Udgaonkar, 2014*). Moreover, the native state has been shown to exist in equilibrium with two high free energy, partially unfolded forms (PUFs), although the time scale of the dynamics that results in the sampling of these PUFs is not known (*Moulick et al., 2015*; *Moulick and Udgaonkar, 2017*).

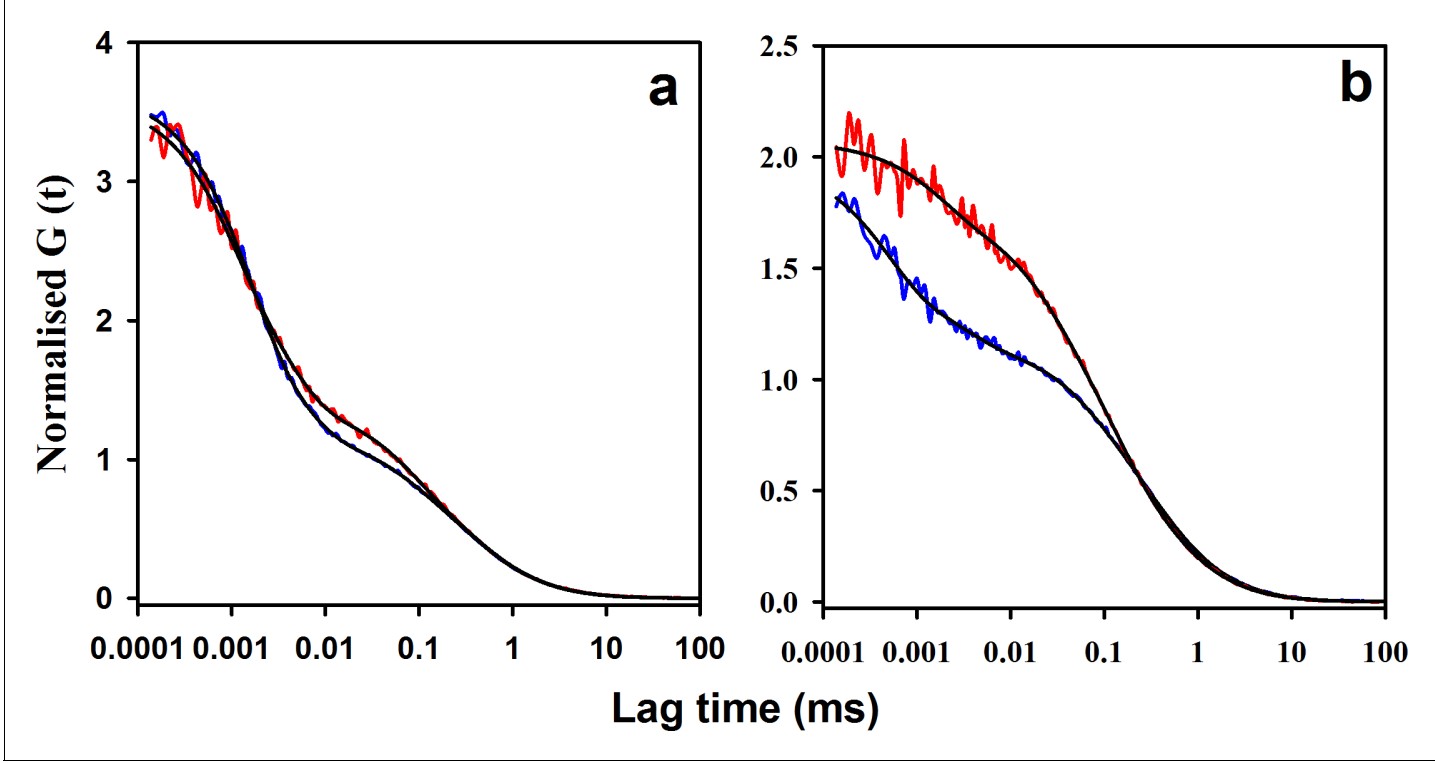

**Figure 5.** Effect of salt on the microsecond dynamics of moPrP at pH 4. (**a**) ACFs of W144/C199-Atto moPrP are shown, in the absence (red line) and in the presence (blue line) of 150 mM NaCl. (**b**) ACFs of W171/C225-Atto moPrP are shown, in the absence (red line) and the presence (blue line) of 150 mM NaCl. The parameters obtained from the fits (black lines) to the data are listed in *Table 1*.

DOI: https://doi.org/10.7554/eLife.44766.019

The following source data and figure supplements are available for figure 5:

**Source data 1.** Effect of salt on the microsecond dynamics of moPrP at pH 4.
DOI: https://doi.org/10.7554/eLife.44766.022
**Figure supplement 1.** Effect of salt on the stability of moPrP.
DOI: https://doi.org/10.7554/eLife.44766.020
**Figure supplement 1—source data 1.** Equilibrium unfolding transitions of W144/C200-Atto moPrP in the presence and absence of salt.
DOI: https://doi.org/10.7554/eLife.44766.021

Several observations suggest that the most likely origin of the faster exponential component observed by PET-FCS is the dynamics within the native state ensemble. (1) There is reasonable quantitative agreement between the values obtained for the fraction of N molecules in non-fluorescent states (N*, N**), determined from the combined amplitude of the two faster exponential components measured by PET-FCS, and from the intensities of the fluorescence spectra of either Atto-labeled protein and its corresponding Trp-less analog (see Results). This validates the assumption that the two faster exponential components arise from native state fluctuations, between fluorescent N and non-fluorescent N* and N**. Moreover, the native state is less fluorescent than the U state for both Atto-labeled proteins, due to the quenching of Atto fluorescence by the nearby Trp-residue in N* and N** within the native state ensemble. Upon unfolding, the quenching is released, and the resulting increase in fluorescence correlated well with the amplitude of the two fast exponential components. (2) The absence of a burst phase change in fluorescence during refolding suggests that the fluorescence of U is similar at high and low denaturant concentrations, ruling out the possibility that a significant fraction of molecules in the U state at high denaturant concentration, exists in the quenched state, as would be the case if the two faster PET-FCS detected fluctuations occur in the U state. (3) When the PET-FCS experiment was carried out at pH 4, where moPrP is destabilized by ~1 kcal/mole, and hence, the U state would be populated more, the amplitude and time constants of the two faster exponential components (*Table 1*) were not significantly different from the values

observed at pH 7 (*Supplementary file 2*). These considerations suggest that the two faster exponential components do not represent the dynamics of the U state. In future, it would be important to validate this interpretation of the two faster exponential components, by measuring conformational fluctuations in the U state by FRET (*Schuler and Hofmann, 2013*).

The possibility that the faster exponential component has contributions from fluctuations in the I state cannot, however, be ruled out at the present time. However, considering the fact that I is substantially destabilized as compared to the native state (*Table 2*), this alternative seems unlikely.

## N* and N** may be dry molten globules within the native state ensemble

In a study of the native state dynamics of a Villin headpiece sub domain, an unlocked conformation was shown to be in equilibrium with N (*Reiner et al., 2010*), by measurements of Triplet Triplet Energy Transfer. This unlocked conformation was likened to a dry molten globular (DMG) intermediate. A DMG has structure very similar to that of N, but with loosened tertiary contacts (*Finkelstein and Shakhnovich, 1989*; *Kiefhaber et al., 1995*; *Jha and Udgaonkar, 2009*; *Sarkar et al., 2013*), and is present on the native side of the major free energy barrier for unfolding. In the current study, the fluctuations seen between N and N*, and between N and N** in the case of W171/C225-Atto moPrP, lead to the separation of helices α2 and α3 in N*, N** in the core of the protein. The fluctuations between N and N* and between N and N** seen in the case of the W144/C199-Atto moPrP, occur on a similar time scale and lead to the separation of the α1-β1-β2 segment from the α2-α3 segment in N*and N**. It is therefore possible N*and N** could therefore be DMG like conformations which are populated within the native state ensemble. N* and N** form on the native side of the free energy barrier and are very dynamic, which might be due to loss of packing interactions in the core of the protein. While N* and N** appear to qualify to be dry molten globules, further validation by structural probes such as FRET is required to confirm whether they are indeed so. It will be important in the future to carry out molecular dynamics simulations to obtain atomistic details about the transitions between N and N* and N**.

## Sub-domain separation and aggregation of prion protein

Aggregation-prone sequences in proteins are normally found sequestered within their cores, resulting in deleterious inter-molecular interactions and consequent aggregation being avoided (*Camilloni et al., 2016*; *De Simone et al., 2011*). Stochastic fluctuations within the native state ensemble can, however, result in the transient exposure of these aggregation-competent patches, and in the initiation of aggregation. In the case of moPrP, it is known that the core of the protein, which comprises of helices α2 and α3, is also the core of the amyloid fibrils (*Singh et al., 2012*). This implies that in order to initiate aggregation, the core of the protein should become exposed to the solvent. Disease-linked mutations of moPrP have been shown to destabilize the native state (*Singh and Udgaonkar, 2015a*), which might result in an increase in the dynamics in the native state. This would be expected to expose aggregation-prone sequence stretches. In the current study, native state fluctuations of moPrP were directly measured, using the PET-FCS methodology. The timescale of such fluctuations was found to be 0.5–20 μs, and is similar to the timescale of native state fluctuations see in the case of a spider silk protein (*Ries et al., 2014*).

The aggregation of the prion protein has been suggested to proceed through sub-domain separation of α1-β1-β2 from the core helices, α2 and α3 (*Singh and Udgaonkar, 2015a*; *Singh and Udgaonkar, 2015b*; *Eghiaian et al., 2007*; *Hafner-Bratkovic et al., 2011*). In the current study, direct experimental evidence for such motions in the native state of the protein has been obtained. The addition of salt is known to initiate aggregation of the prion protein under acidic conditions (*Jain and Udgaonkar, 2010*; *Singh and Udgaonkar, 2016*; *Sengupta et al., 2017*). It has been shown previously that salt destabilizes the human prion protein (*Apetri and Surewicz, 2003*). In the current study, it was observed that salt destabilizes moPrP as well (*Figure 5—figure supplement 1*). The results from the current study suggest that the addition of salt affects the conformational fluctuations occurring within the native state ensemble, which are then expected to result in the sampling of aggregation-competent states. Salt affects both the faster exponential components that appear to originate from fluctuations in the N state which result in the sampling of non-fluorescent N* and N** states. The fluctuations leading to the formation of N** occur in a time domain (~1 μs) where

the data were robust. From the time constants and amplitude of the fast exponential process lead-ing to the formation of N**, the rate constants of formation ($k_{on}$) and dissociation ($k_{off}$) of the pro-ductive dye-Trp complex, N** were determined using *Equations 2 and 3* (*Figure 6*). Quantitative analysis of the fluctuations leading to the formation of N* was not possible because the data were noisy in the time domain of these due to availability of few photons in the time domain of a few hun-dred nanoseconds time scale.

It was observed that the rate constant of the N → N** transition measured for the α2-α3 segment by monitoring fluctuations in W171/C225-Atto moPrP, was nearly an order of magnitude slower than that measured for the α1-α3 segment, by monitoring fluctuations in W144/C199-Atto moPrP (*Figure 6a*). This is likely because of the structural constraints imposed by the disulphide bond pres-ent in moPrP, which staples together the α2-α3 segment, and also by the packing interactions and salt bridges present in the core, which restrict the conformational dynamics in this region. On the other hand, the α1-β1-β2 sub-domain is attached to the core α2-α3 domain through a long flexible loop, which allows rapid structural fluctuations.

The addition of salt had a significant effect on the rate constants of both the N → N** and N** → N transitions measured for the α2-α3 segment, but had a negligible effect on the rate constants measured for the α1-α3 segment (*Figure 6b*). A recent real-time NMR monitored aggregation study of moPrP, has shown that the addition of salt causes the disruption of the salt bridge between resi-dues K193 and E195, which triggers the aggregation process (*Sengupta and Udgaonkar, 2017*). It seems that disruption of this salt bridge by the addition of salt increases flexibility in this region, leading to an increase in the rate constants for fluctuations in the α2-α3 segment. The absence of

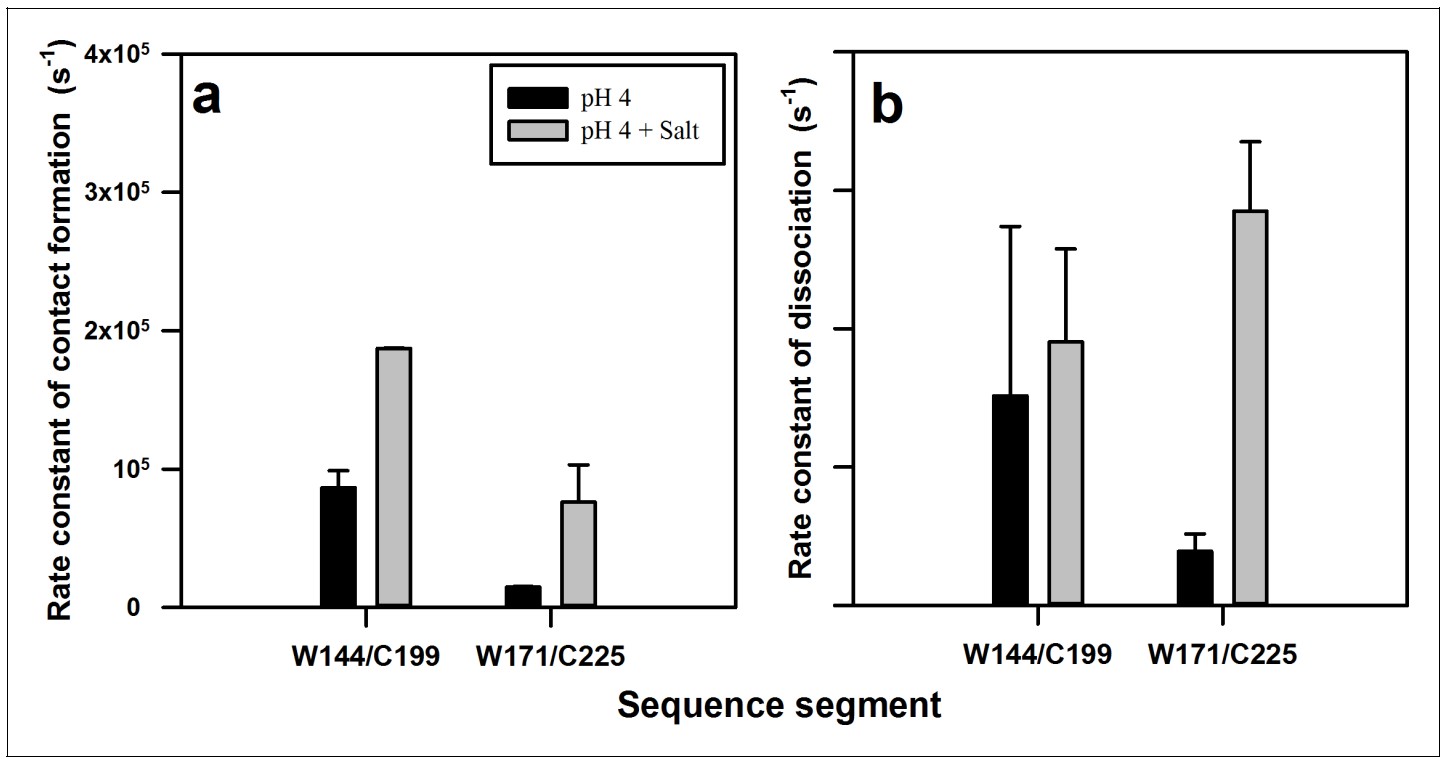

**Figure 6.** Rate constants of contact (complex) formation and dissociation within the native state ensemble of moPrP at pH 4. The rate constants were calculated for the N to N** transition using equilibrium constant $K_2$ and time constant $\tau_2$ obtained from PET FCS ACFs. The rate constants of (**a**) contact formation and (**b**) dissociation for the W144-C199 and W171-C225 segments in the native state, in the absence and the presence of 150 mM NaCl, are shown. Error bars represent the standard deviations obtained from at least two independent repeats of each experiment.
DOI: https://doi.org/10.7554/eLife.44766.024

The following source data is available for figure 6:

**Source data 1.** Rate constants of contact (complex) formation and dissociation within the native state ensemble of moPrP at pH 4.
DOI: https://doi.org/10.7554/eLife.44766.025

the salt bridge would also result in an enhancement of the rate constant of helix separation within the α2-α3 segment, which occurs during the N → N* and N → N** transitions, as there would be fewer ionic interactions to keep the helices together for a long time. Taken all together, the PET-FCS experiments indicate that the addition of salt results in an enhancement in the structural fluctuations in the core of the protein.

Earlier studies have indicated that the dynamics in the U state of prion protein play a role in initiating misfolding and oligomerisation (*Yu et al., 2012*; *Srivastava and Lapidus, 2017*). In the current study it was shown that the fluctuations within the N state of the protein are responsible for the misfolding of the moPrP.

## Role of intermediates in initiating prion protein aggregation

In the case of several proteins, aggregation has been shown to proceed from partially folded intermediates on protein folding/unfolding pathways (*Fink, 1998*; *Hamid Wani and Udgaonkar, 2006*; *Jahn and Parker, 2006*; *De Simone et al., 2011*). For the prion protein too, there is indirect evidence implicating folding intermediates in the aggregation process (*Apetri and Surewicz, 2002*; *Honda et al., 2015*). More recently, a partially unfolded intermediate formed during the unfolding of the CTD of moPrP, was shown directly to be the precursor conformation from which misfolding proceeds (*Moulick and Udgaonkar, 2017*). This intermediate appeared to be similar in structure to the partially unfolded form, PUF2, which is transiently sampled by N. PUF2 had been characterized previously by HX-MS and HX-NMR to have lost much of the structure of N (*Moulick et al., 2015*). At present, it is not known how similar the intermediate I identified in the current study, is to PUF2. The U ↔ I transition is rapid, and I is on the unfolding side of the major free energy barrier for folding (*Figure 7*).

Upon the addition of salt, N is destabilized by 0.7 kcal/mol. I is destabilized less, by 0.4 kcal/mol with respect to the U state (*Table 3* and *Figure 7*), presumably because the K193-E195 salt bridge is absent in it. Thus, the population of I relative to N will be more in the presence of salt. Since misfolding and aggregation are facilitated in the presence of salt, this result suggests that I might play a direct role in misfolding. It seems therefore that both the more unfolded I and the native-like N*, N** are important in the initiation of misfolding and aggregation. The barrier between N and N* as well between N and N** is, however, much smaller than the barrier between N and I (*Figure 7*); hence, N*, N** will be sampled much more frequently than I, *via* fluctuations within the native state ensemble. In future work, it will be important to delineate the relative roles of I, N* and N** in misfolding. It will also be important to determine whether N*and N** form from I on the folding pathway: they are sufficiently similar in structure that contact quenching of the Atto moiety by a Trp residue, occurs in two different structural segments of the protein.

Two earlier studies had suggested that the sampling of conformations that were aggregation-prone, might be driven by fluctuations in the unfolded state of the prion protein. Force spectroscopy studies of the Syrian hamster prion protein (SHaPrP), had suggested that the U state sampled three off-pathway compact intermediates (*Yu et al., 2012*), and that the occupancy of some of these intermediates was more for a disease-linked mutant variant of the protein (*Yu et al., 2012*). It should, however, be noted that the protein used in that study was very different from that used in the current study. The current study was carried out with full length (23-231) moPrP, while the force spectroscopy study had been carried out on truncated SHaPrP (90-231). More importantly, unlike the protein used in the current study, the protein used for the force spectroscopy measurements had a disrupted disulphide bond, and it is known that disruption of the disulphide bond in the prion protein leads to its unfolding to a molten globule form (*Maiti and Surewicz, 2001*; *Honda, 2018*). Fluctuations in SHaPrP (90-231), which could be important in aggregation, had also been identified using Trp-Cys contact quenching experiments carried out at pH 4.4 (*Srivastava and Lapidus, 2017*). In that study, the data appeared to suggest that the fluctuations might arise from the U state, but that could not have been the case because several other studies have shown that SHaPrP (90-231) is natively structured at acidic pH (*Bjorndahl et al., 2011*; *Donne et al., 1997*), including at pH 4 (*Khan et al., 2010*). Hence, aggregation-linked fluctuations of SHaPrP (90-231) need not arise only from the U state. Instead, as suggested by the current study of the dynamics of moPrP, at least some of them are likely to arise from the native state.

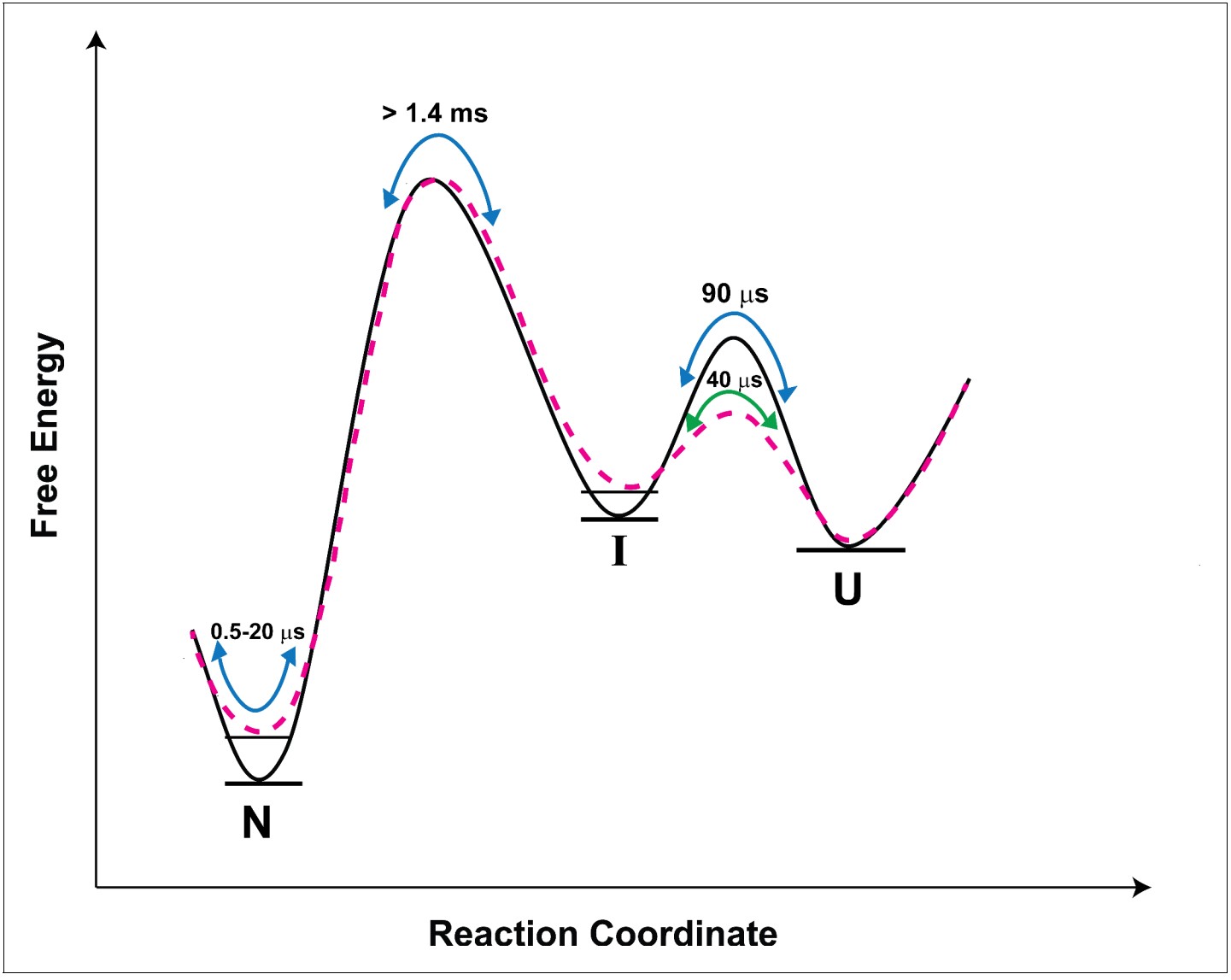

**Figure 7.** Schematic free-energy diagram of moPrP showing the conformational fluctuations in the absence (continuous line) and in the presence of salt (dashed line). The free energy diagram is constructed based on the values of thermodynamic parameters listed in *Table 3*. The U ↔ I transitions are described by the time constants of the slowest exponential process of the ACF. The fluctuations within the native state are represented by the two faster exponential components of the ACF, and occur on the time scale of 0.5 to 20 μs. In the presence of salt, the native state and the I state are destabilized by 0.7, and 0.4 kcal/mol, respectively, with respect to the U state.

DOI: https://doi.org/10.7554/eLife.44766.026

## Conclusion

The study of the native state dynamics of moPrP, across the α1-α3 and α2-α3 segments, monitored by the PET-FCS methodology, using carefully designed single-free cysteine and single-tryptophan-containing variants, reveals three distinct processes, well-separated in time. The faster processes (~0.5–20 μs time constants) represent dynamics within the native state ensemble leading to the formation of the non-fluorescent complexes N* and N**. The slow process (50–100 μs time constant) reports on the U ↔ I transition of folding. The rate constant of the N →N** transition, in the absence of salt, is an order of magnitude slower when measured by monitoring fluctuations across α2-α3, than for the fluctuations across α1-α3, possibly because the fluctuations across α2-α3 are slowed down by the disulphide bond, packing interactions and salt bridges in the core of the protein. The folding landscape and the fluctuations within the native state ensemble are modulated by salt

(*Figure 7*). The addition of salt results in an increase in the rate constants of both the N → N** and N** → N transitions, possibly because critical ionic interactions are lost. The enhancement of the time scale of native state fluctuations upon the addition of salt, suggest that these fluctuations play a role in the initiation of misfolding of the protein.

# Materials and methods

**Key resources table**

| Reagent type (species) or resource | Designation | Source or reference | Identifiers | Additional information |
|---|---|---|---|---|
| Cell line (*E. coli* BL21 Star (DE3)) | *E. coli* BL21 DE3* | Thermo Fisher scientific | | |
| Recombinant DNA reagent (plasmid pET 22b) | Plasmid expressing W144/C199 moPrP | Generated by Dr. Ishita Sengupta (Udgaonkar lab) | | The results of the publication are under communication |
| Recombinant DNA reagent (plasmid pET 22b) | Plasmid expressing Trp-less control protein C199 moPrP | This paper | | This construct was made by site directed mutagenesis |
| Recombinant DNA reagent (plasmid pET 22b) | Plasmid expressing W171/C225 moPrP | This paper | | This construct was made by site directed mutagenesis |
| Recombinant DNA reagent (plasmid pET 22b) | Plasmid expressing Trp-less control protein C225 moPrP | This paper | | This construct was made by site directed mutagenesis |
| Software, algorithm | SigmaPlot | Systat Software Inc | | |
| Software, algorithm | Symphotime-64 | PicoQuant | | |

## Buffers and reagents

All the chemicals used for the study were of the highest purity grade, and were obtained from Sigma. Experiments at pH four were carried out using 20 mM sodium acetate buffer. Experiments at pH seven were carried out using 50 mM phosphate buffer. The buffers contained 0.05% Tween 20 (*Neuweiler et al., 2009*; *Neuweiler et al., 2005*; *Ries et al., 2014*) to prevent adsorption of the protein on to surfaces. All the experiments were carried out at 25°C.

## Protein purification and atto 655 labeling

Full length moPrP contains an unstructured N-terminal region (NTR) and a structured C-terminal domain (CTD) (*Figure 1*). The constructs used in the current study have all the Trp residues mutated to Phe residues, except for a single Trp residue in the CTD at position 144. All the single-cysteine and single-tryptophan containing proteins, as well as the tryptophan-less control proteins, used in the current study contain three cysteine residues, two of which form a disulphide bond in the native protein. These variants were purified using a modified purification protocol for moPrP (*Sengupta and Udgaonkar, 2017*). Briefly, the proteins were expressed in *E. coli* BL21 DE3* cells and purified from inclusion bodies, using Ni-NTA affinity chromatography (*Jain and Udgaonkar, 2008*). The protein was eluted from the Ni-NTA column in 20 mM Tris buffer at pH 8, containing 6 M GdnHCl and 1 mM reduced glutathione (GSH). This was followed by dialysis first against 3 M GdnHCl, and then against 1 M GdnHCl in the same buffer. 0.2 mM oxidized glutathione (GSSG) was added to the protein in 1 M GdnHCl. The final step was dialysis against 20 mM Tris at pH 8. The protein was further purified using cation exchange chromatography, using a CM-FF column (GE Healthcare). The eluate from the ion exchange column was treated with a 5-fold molar excess of Tris (2-carboxyethyl) phosphine hydrochloride to remove the GSH adduct from the free cysteine residue, and then dialyzed against milli-Q water. The protein was flash frozen in liquid nitrogen, and stored at −80°C.

The free cysteine thiol in each variant was labeled with Atto 655 dye by mixing it with a two-fold molar excess of Atto 655 maleimide (ATTO-TEC, GmbH). The labeling reaction was carried out in 50

mM phosphate buffer at pH 7 at 25°C in the dark for 2 hr. The free dye was removed from the reaction mixture by using a HiTrap desalting column (GE Healthcare) or by cation exchange chromatography using HiTrap CM FF column (GE Healthcare), and the protein was buffer-exchanged into milli-Q water, flash frozen, and stored at −80 C. The labeling with Atto 655 resulted in a 650 Da increase in the mass of the protein, which was verified by electrospray ionization mass spectrometry. The concentration of the labeled protein was estimated by measuring the absorbance of the sample at 663 nm, using a molar extinction coefficient of 1,25,000 $M^{-1}$ $cm^{-1}$.

## Equilibrium unfolding experiments

Urea-induced equilibrium unfolding experiments for the Atto 655-labeled moPrP variants, where the fluorescence of the Atto 655-moiety was monitored, were carried out using a FluoroMax-3 (JobinY-von Horiba) spectrofluorometer. The samples were excited at 640 nm, and the fluorescence emission was collected from 645 to 730 nm using an emission bandwidth of 5 nm. The path length of the cuvette used was 1 cm, and the protein concentration used was 10–30 nM.

## PET-FCS experiments

PET-FCS experiments were carried out using a time-resolved confocal microscope, the Micro Time 200 (MT-200) (PicoQuant, Berlin). The light from the 637 nm laser (at 30 µW power) pulsing at 40 MHz was focused on to the sample using a 60 x magnification, 1.2 numerical aperture (NA) objective (Olympus). The fluorescence was collected through the same objective in the epifluorescence mode using a dual band dichroic 480/645 (Chroma). A band pass filter at 690 ± 35 nm was used to eliminate the excitation light, and a 50 µm pinhole was placed in the light path to eliminate out-of-focus light. The fluorescence was split on to two single photon sensitive avalanche photodiodes (MPD-SPAD) using a 50/50 beam splitting cube. The fluorescence from the two SPADs was cross correlated to remove the after-pulsing artifact. Auto correlation functions (ACFs) were generated and fitted to an appropriate model using the Symphotime-64 or SigmaPlot software. The confocal volume was calibrated in each experiment by measuring the ACF of free Atto 655 dye. The concentration of the sample used was 2–20 nM, and the number of molecules in the confocal volume varied from 1 to 8. The cover slips used for the measurements were coated with poly ethylene glycol to suppress surface adsorption of the protein. The ACFs were acquired for 15 min to 1 hr for each sample. The errors reported were from at least two independent measurements. The addition of denaturant resulted in a slowing down of the exponential components, resulting in a partial overlap with the diffusion component. Hence, the experiments with denaturants were carried out using a 100 µm pinhole instead of the 50 µm pinhole. The diffusion time was two-fold longer with the 100 µm pinhole, facilitating reliable measurement of the slower exponential component.

In order to rule out a photo physical phenomenon, such as triplet state formation by the Atto moiety, contributing to the exponential components observed in the PET-FCS experiments, several control experiments were carried out. (a) The excitation power of the laser used was varied between 15 and 60 µW, to determine whether the time constants and amplitudes of the two exponential processes were independent of the laser power or not. (b) The experiments were repeated with the corresponding Trp-less proteins labeled with Atto 655 to determine whether the sub-millisecond exponential components were present or not. (c) The experiments were repeated after an additional buffer exchange step using a 10 kDa centrifugal filter unit (Amicon) which resulted in a further 1000-fold dilution of any free Atto 655 dye present in the sample.

## Microsecond mixing experiments

Sub-millisecond folding experiments were carried out using a custom-built continuous-flow setup, which has been described in detail earlier (*Goluguri and Udgaonkar, 2016*). The sub-millisecond folding experiments were carried out by monitoring the fluorescence of the Atto 655 moiety attached to the protein. The Atto moiety was excited using a Stradus 637 nm cw diode laser from Vortan laser technology. The fluorescence from the mixing chip was collected in the epifluorescence mode using a dual band dichroic 480/645 (Chroma). The fluorescence was collected at 680 nm using a band-pass filter with a band width of 10 nm (Asahi spectra). Microsecond mixing experiments were carried out at two different combined mixing flow rates. The combined flow rate of 7 mL/min

resulted in mixing dead time of 37 µs and total observation time of 1.4 ms. The combined flow rate of 3.5 mL / min resulted in mixing dead time of 75 µs and total observation time of 3 ms.

## Data analysis: Fitting of the ACFs obtained from the PET-FCS measurements

The FCS autocorrelation functions obtained for all proteins were fit to an equation having a diffusion component, as well as three faster chemical reaction components (*Krichevsky and Bonnet, 2002*).

$$G(t) = N^{-1}\left(1 + \frac{t}{\tau_D}\right)^{-1}\left(1 + \frac{t}{w^2\tau_D}\right)^{-\frac{1}{2}}\left(1 + K_1 exp\left(\frac{-t}{\tau_1}\right) + K_2 exp\left(\frac{-t}{\tau_2}\right) + K_3 exp\left(\frac{-t}{\tau_3}\right)\right) \qquad (1)$$

Here $G(t)$ is the auto correlation value, $N$ is the number of molecules in the confocal volume, $t$ is the lag time, $\omega$ is the aspect ratio of the confocal volume, which was calculated to be five for the current system, $\tau_D$ is the diffusion time, $K_1$, $K_2$ and $K_3$ are the amplitudes of the three exponential processes. The corresponding time constants for the fluctuations are represented by $\tau_1$, $\tau_2$ and $\tau_3$. The auto correlation functions obtained for free Atto 655 dye were fitted with an equation having only the diffusion component of *Equation 1*.

The ACFs were normalized to the average number of molecules ($N$) in the confocal volume, obtained from a fit to *Equation 1*. Each ACF was divided with the value $1/N$; this normalization makes the amplitude of the diffusion component equal to 1, and any additional exponential components result in an amplitude above 1.

## Determination of the rate constants of complex formation and dissociation from the ACF

For each of the exponential kinetic phases observed in the ACF, the rate constants of dye-Trp complex/contact formation ($k_{on}$) and dissociation ($k_{off}$) were obtained from the time constant ($\tau$) and amplitude ($K$) of the phase using the following equations

$$1/\tau = k_{on} + k_{off} \qquad (2)$$

$$K = k_{on}/k_{off} \qquad (3)$$

The amplitude reflects the equilibrium constant only when the transition is from the fluorescent to a completely quenched (non-fluorescent) conformation.

## Acknowledgements

We thank members of our laboratory for discussions, and Prof. M K Mathew for discussions and comments on the manuscript. We thank Drs Rahul Roy (IISc, Bengaluru) and GV Pavan Kumar (IISER, Pune) for lending us the 637 nm cw lasers used for the microsecond mixing experiments. JBU is a recipient of a JC Bose National Fellowship from the Government of India.

## Additional information

### Funding

| Funder | Author |
| --- | --- |
| Tata Institute of Fundamental Research | Rama Reddy Goluguri<br>Sreemantee Sen<br>Jayant Udgaonkar |
| Department of Science and Technology, Ministry of Science and Technology | Jayant Udgaonkar |

The funders had no role in study design, data collection and interpretation, or the decision to submit the work for publication.

## Author contributions
Rama Reddy Goluguri, Conceptualization, Data curation, Formal analysis, Investigation, Writing—original draft, Writing—review and editing; Sreemantee Sen, Formal analysis, Investigation; Jayant Udgaonkar, Conceptualization, Resources, Data curation, Formal analysis, Supervision, Funding acquisition, Project administration, Writing—review and editing

## Author ORCIDs
Rama Reddy Goluguri  http://orcid.org/0000-0003-1134-9841
Jayant Udgaonkar  http://orcid.org/0000-0002-7005-224X

## Decision letter and Author response
Decision letter https://doi.org/10.7554/eLife.44766.031
Author response https://doi.org/10.7554/eLife.44766.032

## Additional files

### Supplementary files
• Supplementary file 1. Dependence of the parameters obtained from the PET-FCS ACFs on excitation power. The parameters listed were obtained by fitting the ACFs to *Equation 1* (Materials and Methods). The experiment was carried out using the W144/C199-Atto moPrP variant at pH 7, in the presence of 150 mM salt. The excitation power was measured from the counts from a calibrated photodiode placed before the main dichroic mirror.
DOI: https://doi.org/10.7554/eLife.44766.027

• Supplementary file 2. Parameters obtained from the ACFs of W144/C199-Atto moPrP and W171/C225-Atto moPrP at pH 7.
DOI: https://doi.org/10.7554/eLife.44766.028

• Transparent reporting form
DOI: https://doi.org/10.7554/eLife.44766.029

### Data availability
All data generated during the study are included in the manuscript and supporting files. Source data for Figures 2, 3, 5, 6 and corresponding figure supplements have been uploaded as Excel files.

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
