## [Decision Letter]

Thank you for submitting your article "Microsecond sub-domain motions and the folding and misfolding of the mouse prion protein" for consideration by *eLife*. Your article has been favorably reviewed by three peer reviewers, including Hannes Neuweiler as the Reviewing Editor and Reviewer #1, and the evaluation has been overseen by John Kuriyan as the Senior Editor. The following individuals involved in review of your submission have agreed to reveal their identity: Jan Sykora (Reviewer #3).

The reviewers agree that your manuscript contains new and important kinetic data on near-native state dynamics of the prion protein, which are likely to be involved in the aggregation process of this protein that can ultimately lead to disease. However, the reviewers raise concerns that need to be addressed before publication. Important concerns involve the deduction of molecular-level structural information from kinetic data in general. Specifically, the assignment of conformational states N, N* and N**, the identification of a dry molten globule, and the formation of an on-pathway intermediate appear to be not fully supported by the data presented. Additional concerns involve the nature of the PET fluorescence-quenched state and additional controls that may rule out potential experimental artefacts. The reviews are appended below for your attention.

The reviewers have discussed the reviews with one another and the Reviewing Editor has drafted this decision to help you prepare a revised submission. Please note that the additional experiments requested by reviewer #3 are suggestive and meant to improve the quality of the manuscript but are not mandatory.

*Reviewer #1:*

The manuscript "Microsecond sub-domain motions and the folding and misfolding of the mouse prion protein" by Goluguri et al., reports on the investigation of conformational dynamics of the mouse prion protein (moPrP) using fluorescence spectroscopy. Prion proteins are linked to neurodegenerative diseases and it has been suggested that domain separation within the native protein causes fatal aggregation. Goluguri et al., use photoinduced electron transfer in combination with fluorescence correlation spectroscopy (PET-FCS) to site-specifically probe native-state dynamics of the moPrP from two different sites. They engineer pairs of extrinsic fluorophore and tryptophan (Trp) as PET reporters to probe motions of helices α1/α3 and α2/α3. The authors find sub-millisecond decays in the autocorrelation functions, which they assign to PET fluorescence fluctuations, and which fit to a sum of three exponentials. The slowest ~100-µs kinetic phase is assigned to folding/unfolding transition through comparison with continuous-flow fluorescence kinetic transients. The faster bi-exponential kinetic phase, which is on the 1-20 µs scale, is assigned to native-state fluctuations, which are suggested to reflect transitions to aggregation-prone conformations. These kinetics are modulated by salt, which is known to induce aggregation of the prion protein.

The molecular origins of neurodegenerative disease are a current and important research topic in our aging society. The pathways along which prion proteins misfold and convert to pathological aggregates are not yet explored. Little is known about heterogeneity of the native state of the moPrP, where transiently populated conformations are suggested to be responsible for aggregation. Here, Golguri et al., provide important new insights by measuring sub-millisecond conformational dynamics of the native state of moPrP, which have not been seen before. The PET-FCS data they present exhibit decays of substantial amplitude that originate from conformational motions signalled by engineered PET fluorescence reporters. The dynamics show that native-state heterogeneity is indeed present in the moPrP and involves thermally activated helix motions with populated states that interconvert rapidly.

I have some concerns that should be addressed before publication.

1) The authors state in the Introduction that "addition of salt leads to an enhancement of fluctuations in the core of the protein"; and in the Conclusion "the increase in the amplitude of the native state fluctuations upon addition of salt suggests that they play a role in the initiation of aggregation". But, the amplitudes K1 and K2 measured with and without salt are within error (Table 1). There appears to be hardly an increase of amplitudes with salt, except for amplitude K1 of W171/C225, which increases significantly. The authors should modify their statement and discussion. What is the rational for inferring that conformational fluctuations of moPrP play a role in aggregation from changes in the fluctuation amplitude measured at low and high salt?

2) The authors state (subsection “Structural fluctuations within the native state ensemble give rise to the faster exponential components in the PET-FCS experiments”) that the two fast fluctuations originate from transitions between a fluorescent conformation N and two non-fluorescent conformations N* and N**. But it could also be the other way around. The relaxations might reflect transitions between a non-fluorescent N and two fluorescent N* and N**.

3) How are amplitudes K1 and K2 calculated from steady-state fluorescence spectra? Details are missing. Further: the differences in fluorescence intensity in the spectra should contain K1, K2 and K3. K3 is also an equilibrium transition between a fluorescent and fluorescence-quenched state that contributes to the spectra (albeit originating from folding).

4) On the discussion of the origin of the slow exponential phase in PET-FCS (subsection “The U↔I transition gives rise to the slow exponential component in the PET-FCS experiments”): It may be excluded that these slow fluctuations originate from transitions within the unfolded state because it is hardly populated under the experimental conditions. The authors show evidence that it reflects a U-to-I transition from comparison with continuous-flow data. A fluctuation in N, however, cannot be fully ruled out.

5) The authors subsection "Fluctuations lead to population of a dry molten globule within the native state ensemble". It is problematic to infer structure from kinetics, in general. The observed µs conformational fluctuations do not provide much information on the structure of fluctuating globule.

*Reviewer #2:*

Goluguri et al., investigated the timescale of conformational fluctuations of the mouse prion protein. Since aggregation-related pathologies arise from non-native conformations of the protein, understanding the type, amounts, and timescales at which such non-native states form is of utmost importance for understanding the early stages of the prion disease. The authors used tryptophan quenching (PET) of a well-known reporter dye (Atto 655) in FCS experiments to monitor native state fluctuations. Three processes with relaxation times from 0.5 to 300us were found consistently in two protein variants that map interactions between helices 1 – 3, and 2 – 3, respectively. The experiments were complemented with continuous flow kinetics to identify the global folding/unfolding mechanism and the associated rates. The authors interpret their results in terms of two near-native conformations (N* and N**) that are in equilibrium with the native state (N) and an intermediate (I). Based on plausibility, the authors suggest that (I) forms quickly from the unfolded state, a process that agrees well with the slowest relaxation time (300 us) in the PET-quenching FCS data. The native ensemble (N, N*, N**) is then formed in a slower process (1 ms) from (I). In my opinion, the interpretation sounds plausible and the authors provide good controls to confirm the determined amplitudes in their FCS experiments. However, I think that more evidence is required to proof the suggested mechanism. In the following, I detail a few comments that the authors may want to address before publication is considered:

1) Assigning K3 and τ3 (in FCS) and the fast folding phase to the formation of an on-pathway intermediate is certainly plausible, but not the only explanation of the data. The alternative is the formation of an off-pathway intermediate (I). Can the authors provide further evidence that (I) is indeed on-pathway? The question is particularly relevant because a previous optical tweezer study (Yu et al., (2012)) suggests that PrP forms off-pathway intermediates.

2) The authors stress their assumption that Atto655 is non-fluorescent in the quenched state several times. What is the experimental evidence either from their work or from others that confirms this assumption?

3) One statement confused me. In subsection “Structural fluctuations within the native state ensemble give rise to the faster exponential components in the PET-FCS experiment”, the authors state that values for K1+K2 when calculated from FCS and from fluorescence spectra are similar. The similarity is found for both protein variants (1.8 vs. 2 for W144/C199) and (0.5 vs. 0.7 for W171/C225) as shown in Table 2. Then the authors conclude that the two decay-components 1 and 2 arise from fluctuations across helices α1-α3 and α2-α3. Do the authors want to imply that both distances (α1-α3 and α2-α3) are mapped with each of the variants? If yes, how can this conclusion be drawn from Table 2 that only shows agreement between two methods (FCS and Spectra), but not between variants?

4) Contrary to assumption that the Atto655 fluorescence is always fully quenched in complex with Trp, the authors speculate that the fluorescence is not fully quenched in the I-state (subsection “The U↔I transition gives rise to the slow exponential component in the PET-FCS experiments”). Why? The fact that (I) is only partially structured does not imply that the Atto-Trp complex is different. According to their full-quench hypothesis, only the encounter rate would be affected but not necessarily the fluorescence properties of the complex. This is a problem of consistency (logic) rather than an experimental problem.

5) The authors used 0.05% Tween 20 in their experiments while the critical micelle concentration is 0.007%. Thus, micelle formation may affect the kinetic and thermodynamic properties of the protein. Are the results confirmed at lower Tween 20 concentrations? Other groups typically only use 0.001% Tween 20 in the past ten years.

6) The authors introduce one additional Cys residue in a protein with a disulphide bond. The refolding from IB's therefore requires disulfide shuffling, thus explaining the use of GSH/GSSG. However, have the authors confirmed that the resulting preparations are devoid of mis-folded impurities? Already 5% may cause significant amplitudes in the PET-FCS experiments. Control CD-spectra of the wildtype protein should be compared to those of the labelled and unlabelled spectra obtained for the variants (Figure 1—figure supplement 1).

7) The unfolding transition of W171/C225 does not look very cooperative (Figure 2—figure supplement 3B). To understand the fit and the data better, it would help to indicate the native and unfolded state baseline in these experiments.

8) I would have wished to see a discussion of the results in light of the previous experiments form the Woodside group that seem to conflict with an on-pathway intermediate.

*Reviewer #3:*

In the submitted article, Goluguri et al., studied folding and unfolding of the full-length mouse Prion protein. Although the topic has been in the scope of the scientific community for a longer time and there exists a large amount of the literature data, the approach introduced by the authors brings a certain level of novelty to the field. Specifically, the here-in used methods combining PET quenching with FCS is capable to monitor the conformational fluctuations of the particular protein regions in real time. By means of this method, the authors identified the time-scales of the folding dynamics of PrP spanning the microsecond to submillisecond time-scale, which brings novel information on the PrP behavior. Another strong part of the work is a careful design of the experimental system. The mutations inserted into the wild-type proteins were well thought over, they address the interesting protein regions and fulfill all the pre-requisites for the successful implementation of PET-FCS experiments. Moreover, conformations of the two PrP variants do not seem to be affected by the mutations. The interpretation and evaluation of the PET-FCS experiments (including the salt data), providing the insight into the microsecond protein dynamics, are carried out in a satisfactory manner.

In contrast, the stop-flow experiments appear to be more questionable. As correctly stated by the authors urea disrupts the quenching of the Atto655 by Trp. This does not hold true only in the case of PET-FCS, but also for the PET-stop flow experiments. Therefore, the folding and unfolding rates deduced from the stop flow experiments are likely to be the combination of the folding/unfolding dynamics and destabilization of the dye-Trp complex by urea. Moreover, the applied model comprising I <-> U transition does not seem to be valid for mutant W171/C225-Atto moPrP, which show rather complex behavior (Figure 2—figure supplement 3B).

Another weak point of the manuscript seems to be the over-interpretation of the collected data. For instance, there is no evidence given that N-state represent the fluorescent form, while N* and N** are non-fluorescent ones. The character of the N* and N** states as the dry molten globule needs further justification. The fact that PET-FCS rate constants overlap with the fast constants of the stop-flow measurements do not automatically imply that the same conformational motions are followed by these different approaches. Without additional methods (for instance molecular dynamics simulations), it is speculative to draw the conclusions on the atomistic level as in section "The two faster PET-FCS fluctuations likely represent native state fluctuations".

In conclusion, the article brings novel findings on the fast microsecond dynamics of the mouse PrP protein, which might be of interest. However, the article in the present form suffers from the potential artifacts on the interpretation of the folding/unfolding mechanisms on the longer sub-millisecond time scale. I recommend focusing more on the fast components revealed by PET-FCS, formulating the Results section and Discussion section in more concise and less speculative manner and to perform additional experiments specified below. After the major revision the article could be accepted for publishing in *eLife*.

To obtain the deeper insight into the longer sub-millisecond folding/unfolding of the mouse PrP the following data shall be given:

- To measure the unfolding/refolding kinetics for more chaotropes.

- To measure the CD spectra of the partially unfolded and fully unfolded forms of both W144/C199 and W171/C225 variants (i.e. at different chaotrope concentrations).

- To present the refolding kinetic trace, effect of salt on the stability, and ACF in 6M urea at pH 4 also for W171/C225-Atto moPrP.

---

## [Author Response]

Reviewer #1:[…] 1) The authors state in the Introduction that "addition of salt leads to an enhancement of fluctuations in the core of the protein"; and in the Conclusion "the increase in the amplitude of the native state fluctuations upon addition of salt suggests that they play a role in the initiation of aggregation". But, the amplitudes K1 and K2 measured with and without salt are within error (Table 1). There appears to be hardly an increase of amplitudes with salt, except for amplitude K1 of W171/C225, which increases significantly. The authors should modify their statement and discussion. What is the rational for inferring that conformational fluctuations of moPrP play a role in aggregation from changes in the fluctuation amplitude measured at low and high salt?

The reviewer is correct: the amplitudes of the fast fluctuations are hardly affected by salt. The time scale of fluctuations, is however, affected. We have now modified the sentences in the Introduction and Conclusion section, to clarify that the time scale of fluctuations in the core of the protein is enhanced in the presence of salt. The rationale for inferring that conformational fluctuations in the N state of moPrP play a role in initiation of misfolding is the observation that the time scale of fluctuations became faster upon addition of salt, which is also evident from Figure 6. We no longer infer this from the amplitudes.

2) The authors state (subsection “Structural fluctuations within the native state ensemble give rise to the faster exponential components in the PET-FCS experiments”) that the two fast fluctuations originate from transitions between a fluorescent conformation N and two non-fluorescent conformations N* and N**. But it could also be the other way around. The relaxations might reflect transitions between a non-fluorescent N and two fluorescent N* and N**.

We agree with the reviewer that the two fast fluctuations of the N state could also arise from sampling of two fluorescent N states by a non-fluorescent N state. We have included this possibility in the revised manuscript. It should, however, be noted that this alternative possibility will not affect any of our results and the conclusion drawn from them. in subsection “Structural fluctuations within the native state ensemble give rise to the faster exponential components in the PET-FCS experiments” we now state “It is also possible that, instead, N is non-fluorescent and that N* and N** are fluorescent. The current study cannot distinguish between these possibilities, and the model with one fluorescent N state and the two non-fluorescent N* and N** states, has been chosen only because it seems to be the simpler alternative. It should be noted that the alternative possibility does not affect any of the conclusions drawn in the current study. In both cases the fluctuations arise from N sampling N* and N** in the native state ensemble.”

3) How are amplitudes K1 and K2 calculated from steady-state fluorescence spectra? Details are missing. Further: the differences in fluorescence intensity in the spectra should contain K1, K2 and K3. K3 is also an equilibrium transition between a fluorescent and fluorescence-quenched state that contributes to the spectra (albeit originating from folding).

The procedure is described in the footnote of Table 2. K3 is not included because U and I are hardly populated under the native conditions in which the spectra shown in Figure 2 were acquired. We realize that an error in the legend to Figure 2 (now corrected) may have confused the reviewer.

4) On the discussion of the origin of the slow exponential phase in PET-FCS (subsection “The U↔I transition gives rise to the slow exponential component in the PET-FCS experiments”): It may be excluded that these slow fluctuations originate from transitions within the unfolded state because it is hardly populated under the experimental conditions. The authors show evidence that it reflects a U-to-I transition from comparison with continuous-flow data. A fluctuation in N, however, cannot be fully ruled out.

The reviewer is correct. In subsection “The U↔I transition gives rise to the slow exponential component in the PET-FCS experiments”, we now state: “Another explanation is that it represents slow dynamics in the native ensemble, or perhaps even the sampling of a partially unfolded intermediate by N. This explanation cannot be fully ruled out.”

5) The authors subsection "Fluctuations lead to population of a dry molten globule within the native state ensemble". It is problematic to infer structure from kinetics, in general. The observed µs conformational fluctuations do not provide much information on the structure of fluctuating globule.

We agree with the reviewer, and in fact, state in subsection “N* and N** may be dry molten globules within the native state ensemble” that “while N* and N** appear to qualify to be dry molten globules, further validation by structural probes such as FRET is required to confirm whether they are indeed so.” We realize, however, that our section heading, quoted by the reviewer was misleading and we have changed it suitably. We have decreased the emphasis on N* and N** being dry molten globule-like and we have removed this claim from the abstract and conclusions of the revised manuscript.

Reviewer #2:[…] 1) Assigning K3 and τ3 (in FCS) and the fast folding phase to the formation of an on-pathway intermediate is certainly plausible, but not the only explanation of the data. The alternative is the formation of an off-pathway intermediate (I). Can the authors provide further evidence that (I) is indeed on-pathway? The question is particularly relevant because a previous optical tweezer study (Yu et al., (2012)) suggests that PrP forms off-pathway intermediates.

The reviewer is correct, and in subsection “The U↔I transition gives rise to the slow exponential component in the PET-FCS experiments” we now state: “It should be noted that the possibility that I is, instead, an off-pathway intermediate cannot be ruled out, even though several previous kinetic studies of the folding of the mouse prion protein (R. P. Honda et al., 2015; Moulick, Goluguri and Udgaonkar, 2019) and of other mammalian prion proteins (Apetri et al., 2006; Chen et al., 2011) have implicated an on-pathway intermediate being formed on a time scale similar to that observed in this study. It should be noted that the observation that I like N is destabilized upon the addition of salt, suggests that the salt-perturbed native interaction is also present in I, which is also consistent with I being populated on the pathway from U to N.”

We have now included a paragraph comparing the results of the current study to the results of the previous optical tweezer study (Yu et al., 2012) and Trp-Cys quenching study (Srivastava and Lapidus, 2017): In the subsection “Role of intermediates in initiating prion protein aggregation”, we now state: “Two earlier studies had suggested that the sampling of conformations that were aggregation-prone, might be driven by fluctuations in the unfolded state of the prion protein. […] Hence, aggregation-linked fluctuations of SHaPrP (90-231) need not arise only from the U state. Instead, as suggested by the current study of the dynamics of moPrP, at least some of them are likely to arise from the native state.”

Hence, the results of the current PET-FCS study are not in contradiction of the results of the previous studies on prion protein that probed the dynamics of the unfolded state.

2) The authors stress their assumption that Atto655 is non-fluorescent in the quenched state several times. What is the experimental evidence either from their work or from others that confirms this assumption?

In a previous study of the quenching of organic dyes MR121 and Atto 655 by tryptophan it was shown that the dye forms a non-fluorescent complex with tryptophan by comparing the equilibrium constant obtained from FCS experiments with the equilibrium constants obtained from steady state fluorescence spectra (Doose et al., 2005). In the subsection “The two faster PET 428 -FCS fluctuations likely represent native state fluctuations”, we explain how our results are results are consistent with the dye being fully quenched in N* and N**. We state: “There is reasonable quantitative agreement between the values obtained for the fraction of N molecules in non-fluorescent states (N*, N**), determined from the combined amplitude of the two faster exponential components measured by PET-FCS, and from the intensities of the fluorescence spectra of either Atto-labeled protein and its corresponding Trp-less analog (see Results section). This validates the assumption that the two faster exponential components arise from native state fluctuations, between fluorescent N and non-fluorescent N* and N.”

3) One statement confused me. In subsection “Structural fluctuations within the native state ensemble give rise to the faster exponential components in the PET-FCS experiment”, the authors state that values for K1+K2 when calculated from FCS and from fluorescence spectra are similar. The similarity is found for both protein variants (1.8 vs. 2 for W144/C199) and (0.5 vs. 0.7 for W171/C225) as shown in Table 2. Then the authors conclude that the two decay-components 1 and 2 arise from fluctuations across helices α1-α3 and α2-α3. Do the authors want to imply that both distances (α1-α3 and α2-α3) are mapped with each of the variants? If yes, how can this conclusion be drawn from Table 2 that only shows agreement between two methods (FCS and Spectra), but not between variants?

We realize that our construction of the sentence was misleading. We did not want to imply that both distances are mapped with each of the variants. Our conclusion, as we now state in subsection “Structural fluctuations within the native state ensemble give rise to the faster exponential components in the PET-FCS experiments” is only: This similarity suggested that the two faster exponential components in the ACFs arise from structural fluctuations within the native state ensemble of moPrP.

4) Contrary to assumption that the Atto655 fluorescence is always fully quenched in complex with Trp, the authors speculate that the fluorescence is not fully quenched in the I-state (subsection “The U↔I transition gives rise to the slow exponential component in the PET-FCS experiments”). Why? The fact that (I) is only partially structured does not imply that the Atto-Trp complex is different. According to their full-quench hypothesis, only the encounter rate would be affected but not necessarily the fluorescence properties of the complex. This is a problem of consistency (logic) rather than an experimental problem.

We realize that it was not right to imply that the Atto-dye complex in I is different from that in the N ensemble. Comparison of data from PET FCS and ensemble fluorescence measurements indicated that the fluorescence of Atto moiety is completely quenched in N state of moPrP (see response to query#2). We do not have such an experimental evidence for the extent of quenching happening in I state. We now state in subsection “The U↔I transition gives rise to the slow exponential component in the PET-FCS experiments”: “It is not possible, at the present time, to be certain that the fluorescence of the Atto moiety is completely quenched by the Trp residue, in I. There is, however, no data to support the dye-Trp complex being different in I than in the N ensemble. If complete quenching does not occur in the dye-Trp complex, then the amplitude of the slowest exponential process observed by PET-FCS would not correspond to the true equilibrium constant for the pre-equilibrium established between U and I during the fast kinetic phase of folding.”

5) The authors used 0.05% Tween 20 in their experiments while the critical micelle concentration is 0.007%. Thus, micelle formation may affect the kinetic and thermodynamic properties of the protein. Are the results confirmed at lower Tween 20 concentrations? Other groups typically only use 0.001% Tween 20 in the past ten years.

As we now state in subsection “Protein purification and Atto 655 labeling”, we have used the same Tween 20 concentration as used in other studies of protein dynamics that utilized PET-FCS (Daidone et al., 2010; Doose et al., 2005; Jensen et al., 2011; Neuweiler et al., 2010; Neuweiler et al., 2005; Neuweiler et al., 2009; Ries et al., 2014; Schulze et al., 2016). While this concentration is above the CMC, our results indicate that the protein does not interact with micelles: the hydrodynamic radius determined for the protein is the same as measured previously by DLS in the absence of any Tween 20 (see subsection “moPrP shows three reaction kinetic phases in the sub-millisecond time scale”).

6) The authors introduce one additional Cys residue in a protein with a disulphide bond. The refolding from IB's therefore requires disulfide shuffling, thus explaining the use of GSH/GSSG. However, have the authors confirmed that the resulting preparations are devoid of mis-folded impurities? Already 5% may cause significant amplitudes in the PET-FCS experiments. Control CD-spectra of the wildtype protein should be compared to those of the labelled and unlabelled spectra obtained for the variants (Figure 1—figure supplement 1).

We have now included the CD spectrum of the wildtype protein is Figure 1—figure supplement 1. The spectrum of the wild type protein looks very similar to those of the mutant variants used in the current study.

7. The unfolding transition of W171/C225 does not look very cooperative (Figure 2—figure supplement 3B). To understand the fit and the data better, it would help to indicate the native and unfolded state baseline in these experiments.

Figure 2—figure supplement 3 has been modified to indicate the native and unfolded state baselines.

8. I would have wished to see a discussion of the results in light of the previous experiments form the Woodside group that seem to conflict with an on-pathway intermediate.

We have now included a paragraph in subsection “Role of intermediates in initiating prion protein aggregation” which states: “Two earlier studies had suggested that the sampling of conformations that were aggregation-prone, might be driven by fluctuations in the unfolded state of the prion protein. […] Hence, aggregation-linked fluctuations of SHaPrP (90-231) need not arise only from the U state. Instead, as suggested by the current study of the dynamics of moPrP, at least some of them are likely to arise from the native state.”

Reviewer #3:In the submitted article, Goluguri et al., studied folding and unfolding of the full-length mouse Prion protein. Although the topic has been in the scope of the scientific community for a longer time and there exists a large amount of the literature data, the approach introduced by the authors brings a certain level of novelty to the field. Specifically, the here-in used methods combining PET quenching with FCS is capable to monitor the conformational fluctuations of the particular protein regions in real time. By means of this method, the authors identified the time-scales of the folding dynamics of PrP spanning the microsecond to submillisecond time-scale, which brings novel information on the PrP behavior. Another strong part of the work is a careful design of the experimental system. The mutations inserted into the wild-type proteins were well thought over, they address the interesting protein regions and fulfill all the pre-requisites for the successful implementation of PET-FCS experiments. Moreover, conformations of the two PrP variants do not seem to be affected by the mutations. The interpretation and evaluation of the PET-FCS experiments (including the salt data), providing the insight into the microsecond protein dynamics, are carried out in a satisfactory manner.In contrast, the stop-flow experiments appear to be more questionable. As correctly stated by the authors urea disrupts the quenching of the Atto655 by Trp. This does not hold true only in the case of PET-FCS, but also for the PET-stop flow experiments. Therefore, the folding and unfolding rates deduced from the stop flow experiments are likely to be the combination of the folding/unfolding dynamics and destabilization of the dye-Trp complex by urea. Moreover, the applied model comprising I <-> U transition does not seem to be valid for mutant W171/C225-Atto moPrP, which show rather complex behavior (Figure 2—figure supplement 3B).

The reviewer suggests that the measured rate constant for the formation of I may not be the true rate constant of folding, because urea will interfere with dye-Trp complex formation which occurs as a consequence of folding. This is certainly possible, but there is data suggesting that this not happen. As we now state in subsection “The U↔I transition gives rise to the slow exponential component in the PET-FCS experiments”; “It is also seen that the rate constant of the slow exponential process measured in 0.75 M urea by PET FCS, matches the fast rate constant of folding in 0.75 M urea. It is important to note that the fast rate constants of folding in 1.25 to 2.5 M urea, where the quenching efficiency is suppressed, extrapolate to the rate constant of folding in 0.75 M urea (Figure 3D), where the quenching efficiency is unaffected by urea. This observation indicates that any suppression of the PET quenching efficiency that occurs at the high urea concentrations does not significantly affect the time courses of folding at those urea concentrations.”

It is true that W171/C225 Atto moPrP exhibits an unusual urea-induced equilibrium unfolding curve. Nevertheless, given that the time constant and amplitude of the slowest exponential component observed in the ACF of this protein are similar to those observed for W144/C199 Atto moPrP, it is expected that the slow exponential ACF component seen for W171/C225 Atto moPrP also represents the folding of U to I. In future studies, it will be important to test whether this is true. Such studies are also expected to provide an understanding of the complex urea-induced equilibrium unfolding curve of W171/C225 Atto moPrP.

Another weak point of the manuscript seems to be the over-interpretation of the collected data. For instance, there is no evidence given that N-state represent the fluorescent form, while N* and N** are non-fluorescent ones. The character of the N* and N** states as the dry molten globule needs further justification. The fact that PET-FCS rate constants overlap with the fast constants of the stop-flow measurements do not automatically imply that the same conformational motions are followed by these different approaches. Without additional methods (for instance molecular dynamics simulations), it is speculative to draw the conclusions on the atomistic level as in section "The two faster PET-FCS fluctuations likely represent native state fluctuations".

We now address these concerns.

In subsection “Structural fluctuations within the native state ensemble give rise to the faster exponential components in the PET-FCS experiments”, we now state: “It is also possible that N is non-fluorescent and that N* and N** are fluorescent. The current study cannot distinguish between these possibilities, and the model with one fluorescent N state and the two non-fluorescent N* and N** states, has been chosen only because it seems to be the simpler alternative. It should be noted that the alternative possibility does not affect any of the conclusions drawn in the current study. In both cases the fluctuations arise from N sampling N* and N** in the native state ensemble.”

We realize that in the absence of definitive evidence, it was wrong to so strongly posit that N* and N** are dry molten globules. We have toned down the text, and state in subsection “N* and N** may be dry molten globules within the native state ensemble” that “while N* and N** appear to qualify to be dry molten globules, further validation by structural probes such as FRET is required to confirm whether they are indeed so.” We also realize that our section heading, quoted by the reviewer was misleading, and we have changed it suitably. We have decreased the emphasis on N* and N** being dry molten globule-like, and we have removed this claim from the Abstract and Conclusion section of the revised manuscript.

We believe we have provided strong support for our conclusion that the slowest exponential component observed in the PET FCS experiments and that the fast phase observed in the folding experiments represent the same conformational motions. Not only are the rate constants overlapping, but their dependences on urea concentration, are the same.

In the subsection "The two faster PET-FCS fluctuations likely represent native state fluctuations", we have explained in detail why the experimental data make the conclusion likely. Molecular dynamics simulations are outside the scope of the current study, which has combined (rarely done previously) both ensemble and single molecule measurements of folding, and in subsection “N* and N** may be dry molten globules within the native state ensemble”, we now state:” It will be important in the future to carry out molecular dynamics simulations to obtain atomistic details about the transitions between N and N* and N**.”

In conclusion, the article brings novel findings on the fast microsecond dynamics of the mouse PrP protein, which might be of interest. However, the article in the present form suffers from the potential artifacts on the interpretation of the folding/unfolding mechanisms on the longer sub-millisecond time scale. I recommend focusing more on the fast components revealed by PET-FCS, formulating the results and discussion sections in more concise and less speculative manner and to perform additional experiments specified below. After the major revision the article could be accepted for publishing in eLife.To obtain the deeper insight into the longer sub-millisecond folding/unfolding of the mouse PrP the following data shall be given:

- To measure the unfolding/refolding kinetics for more chaotropes.

It is known that moPrP forms oligomers in the presence of guanidine hydrochloride at pH 4 because of the ionic nature of the chaotrope. This was one of the reasons we chose to perform the current study with urea.

- To measure the CD spectra of the partially unfolded and fully unfolded forms of both W144/C199 and W171/C225 variants (i.e. at different chaotrope concentrations).

It is not clear what this data would add to the considerable amount of data already present in the manuscript. These measurements appear to be outside the scope of the current study.

*- To present the refolding kinetic trace, effect of salt on the stability, and ACF in 6M urea at pH 4 also for W171/C225-Atto moPrP*.

As we state in response to the first point of the reviewer this will be part of our future work. This will require a large amount of protein for the microsecond mixing experiments because of the small change observed during folding.